# TWISTED: ENHANCING TRANSFORMER WORLD MODELS WITH SPATIO-TEMPORAL ENCODING AND GRAPH-BASED OPTIMAL DECODING

## ABSTRACT

Model-based reinforcement learning improves sample efficiency by using learned world models to simulate experiences for training agents. Recent world models that leverage transformers demonstrate high quality simulations, leading to better agent performance. However, transformer world models underutilize spatial relationships between visually adjacent tokens, which are critical when interacting in visual environments. Additionally, current models rely on sampling methods for transformer decoding that do not leverage visual similarities among subsequent frames. To address these limitations, we introduce TWISTED, a transformer world model with 3D spatio-temporal positional encoding and a graph-based optimal decoding strategy specific to 2D visual environments. Our experiments show state-of-the-art performance on the Craftax-classic, Craftax, MinAtar, and Atari 100K benchmarks, challenging visual environments requiring long-horizon object recall and interaction. The proposed method achieves a *return* of 72.5% and a *score* of 35.6% on Craftax-classic, significantly surpassing the previous best of 67.4% and 27.9%. We plan to release our source code on GitHub upon acceptance.

## 1 INTRODUCTION

Reinforcement learning (RL) provides a framework for training agents to interact with their environment through reward signals (Sutton & Barto, 2018). To avoid heavy reliance on costly environment interactions, model-based RL learns a predictive model of the environment dynamics, enabling the agent to simulate future trajectories called "imaginations" (Hafner et al., 2023; Micheli et al., 2022). Recently, transformers have emerged as powerful world models (Micheli et al., 2022; Dedieu et al., 2025). They treat sequences of past states and actions as token streams and predict the next state token-by-token. However, adapting transformers to world modeling faces challenges that have not been fully explored. First, transformers typically rely on one-dimensional positional encodings to capture token order, which may be insufficient for visual environments. Second, effective token decoding strategies tailored to world models have yet to be investigated.

The standard choice for positional encoding is Rotary Position Embedding (RoPE), which captures the relative distance between tokens (Su et al., 2024). While effective in text domains, RoPE is less suited to visual environments, where data is naturally structured in three dimensions—two spatial (within frames) and one temporal (across frames). RoPE encodes only one-dimensional relationships, leading to a loss of fine-grained spatio-temporal structure when applied naively to vision-based tasks.

For transformer decoding, common practice is to sample tokens from the output probabilities in parallel or sequentially (Micheli et al., 2022; Dedieu et al., 2025). Parallel decoding, while efficient, ignores dependencies between output tokens, often leading to hallucinations in complex environments such as duplicated objects. On the other hand, sequential decoding incurs higher computational costs and has been shown to degrade generation quality due to auto-regressive drift (Dedieu et al., 2025). Moreover, neither decoding scheme leverages the fact that sequential frames in visual environments are highly similar (see Figure 1(a)). Meanwhile, exploiting similarities between images has been studied extensively in computer vision, leading to the development of optical flow-based techniques. (Brox et al., 2004; Vedula et al., 2005; Perazzi et al., 2016).

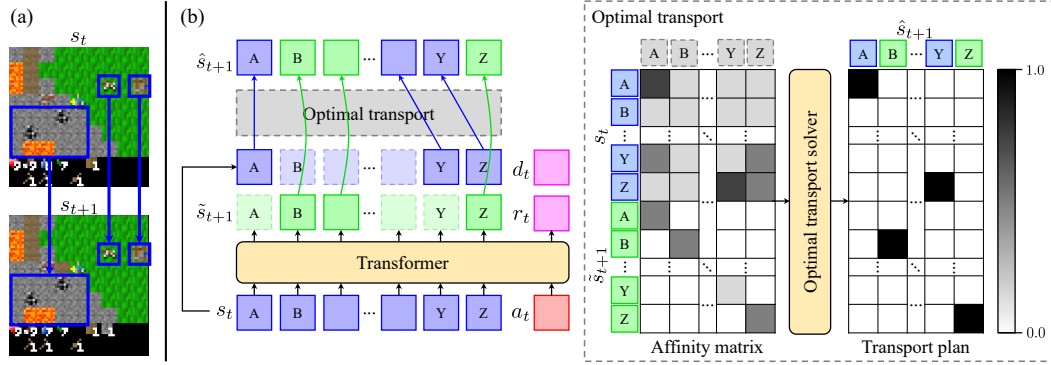

Figure 1: (a) Sequential frames in visual environments like Craftax-classic preserve most of the information from the previous frame. (b) Our proposed world model enhances next state prediction by solving an optimal transport problem with previous state tokens ($s_t$, blue) and the transformer's output for candidate next-state tokens ($\tilde{s}_{t+1}$, green) to generate the final next-state tokens ($\hat{s}_{t+1}$). Optimal transport defines an affinity matrix from the $s_t$ and $\tilde{s}_{t+1}$ tokens to the positions for $\hat{s}_{t+1}$. A solver takes the affinity matrix and produces a transport plan, assigning a token from $s_t$ or $\tilde{s}_{t+1}$ to each final next-state token in $\hat{s}_{t+1}$. This approach enables effective reuse of relevant past tokens.

In this regard, we propose TWISTED (Transformer World model with Informed Spatio-Temporal Encoding and Decoding), a transformer world model designed for 2D visual RL environments. TWISTED introduces two key innovations:

1. A 3D spatio-temporal positional encoding that combines absolute and relative encodings across space and time, preserving both spatial structure and temporal structure.

2. A graph-based optimal decoding scheme grounded in optimal transport, which formulates decoding as a transport problem between the previous frame's optimally decoded tokens and the transformer's predictions for next tokens. This enables partial reuse of previous tokens, reducing hallucinations and improving object persistence over time.

We evaluate TWISTED on the Craftax-classic, Craftax, MinAtar, and Atari 100K benchmarks. Craftax-classic is a challenging 2D open-world game featuring long-horizon tasks and dynamic enemies (Matthews et al., 2024). TWISTED achieves a return of 72.5% and a score of 35.6%, setting a new state-of-the-art and outperforming the previous best results of 67.4% and 27.9%, respectively (Dedieu et al., 2025). Craftax is a harder environment based on Craftax-classic, in which TWISTED also exceeds baselines (Matthews et al., 2024). MinAtar is a suite of 4 Atari games with simplified representations, which tests generality across different game dynamics (Young & Tian, 2019). TWISTED surpasses the previous state of the art for model-based RL in all 4 games (Dedieu et al., 2025). Atari 100K is a suite of 26 Atari games with diverse visual structure. TWISTED surpasses the previous state-of-the-art token-based world model (Cohen et al., 2025) across the 26 games.

## 2 PRELIMINARIES

### 2.1 MODEL-BASED REINFORCEMENT LEARNING

Reinforcement learning considers a Partially Observable Markov Decision Process, characterized by $(\mathbb{S}, \mathbb{A}, \Omega, T, O, R, \gamma)$, where $\mathbb{S}$ is a set of states, $\mathbb{A}$ is a set of discrete actions, $\Omega$ is a set of observations, $T$ gives the transition probabilities between states $T(s' \mid s, a)$, $O$ gives the observation probabilities $O(o \mid s)$, and $R$ is a reward function $R(s, a)$ (Sutton & Barto, 2018). The goal is to find a policy $\pi$ which chooses actions for each state that maximizes the expected discounted return $\mathbb{E}_\pi \left[ \sum_{t \geq 0} \gamma^t r_t \right]$, where $\gamma$ is a discount factor. A world model takes an input of previous state $s_t$ and action $a_t$, then returns a predicted output of next state $\hat{s}_{t+1}$, reward $r_t$, and done signal $d_t$, similar to the real environment. The agent collects real environment trajectories during training by interacting

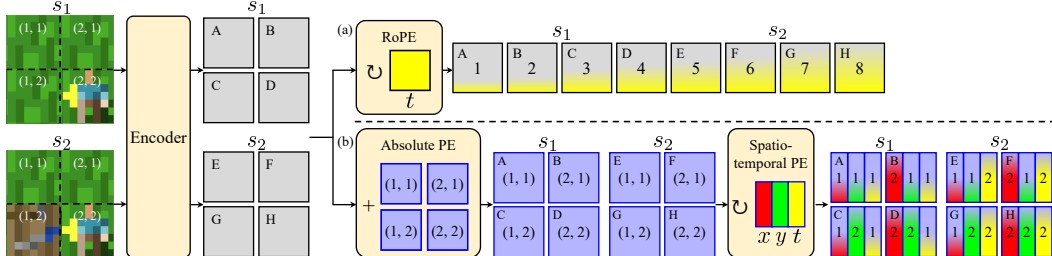

Figure 2: Use of positional encoding in (a) original RoPE vs. (b) our method. Original RoPE makes vertically adjacent (A and C) and temporally adjacent (A and E) tokens far from each other, while our method brings them together along two out of three axes. Moreover, original RoPE makes spatially distant tokens (D and E) adjacent, while our method separates them across all axes.

with the environment using the policy $\pi$. Then the world model trains on the trajectories saved in the replay buffer. Over the course of training, the agent is trained on both the trajectories collected from the real environment and generated trajectories from the world model, called *imaginations*.

## 2.2 RoPE

Rotary Position Embedding (RoPE) is a positional encoding method that injects positional information into a transformer's attention mechanism by applying rotations to query and key vectors (Su et al., 2024). These rotations cause the attention operation to naturally encode relative offsets between tokens. Concretely, each input token embedding is partitioned into pairs of coordinates, with each pair forming a 2D subspace where a rotation is applied according to the token's 1D position index. Owing to its simplicity and scalability, RoPE has become the standard positional encoding in modern transformer architectures.

## 2.3 Optimal transport

Optimal transport is a family of optimization problems that compares and aligns probability distributions based on a given cost of moving mass between elements (Peyré & Cuturi, 2019). Optimal transport considers probability distributions $\mathbf{a} \in \Delta^{n-1}$ and $\mathbf{b} \in \Delta^{m-1}$ over the source and target domains, respectively. It seeks a transport plan $\mathbf{\Pi} \in \mathbb{R}_{+}^{n \times m}$ that minimizes the cost $\langle \mathbf{\Pi}, \mathbf{C} \rangle = \sum_{i=1}^{n} \sum_{j=1}^{m} \Pi_{ij} C_{ij}$, subject to the marginal constraints $\mathbf{\Pi} \mathbf{1}_m = \mathbf{a}$ and $\mathbf{\Pi}^\top \mathbf{1}_n = \mathbf{b}$.

To solve optimal transport problems efficiently, regularized variants of optimal transport have been proposed. One popular approach introduces an entropic regularization term to the objective, leading to the *Sinkhorn distance*, which can be computed efficiently using iterative matrix scaling (Cuturi, 2013). The Sinkhorn algorithm solves the regularized problem in $O(n^2/\epsilon^2)$ time for a desired approximation error $\epsilon$, making it practical for large-scale problems.

## 2.4 Tokenizer

Transformer world models require a tokenizer to convert states and actions into discrete tokens for the transformer. Dedieu et al. (2025) introduced a tokenizer that converts the visual observation to tokens using nearest neighbor patch lookup. Each token represents a particular visual patch of the image state. First, each frame is divided into a grid of $L$ visual patches $\{p_1, \ldots, p_L\}$, where $p_i \in [0,1]^{h \times w \times 3}$ with height $h$ and width $w$. The tokenizer maintains a codebook $C = \{c_1, \ldots, c_K\}$, consisting of $K$ codes $c_i \in [0,1]^{h \times w \times 3}$. Each patch $p$ is mapped to a token $q$ by finding its nearest neighbor in the codebook:

$$q = \text{enc}(p) = \underset{1 \leq i \leq K}{\arg\min} \|p - c_i\|_2^2 .$$

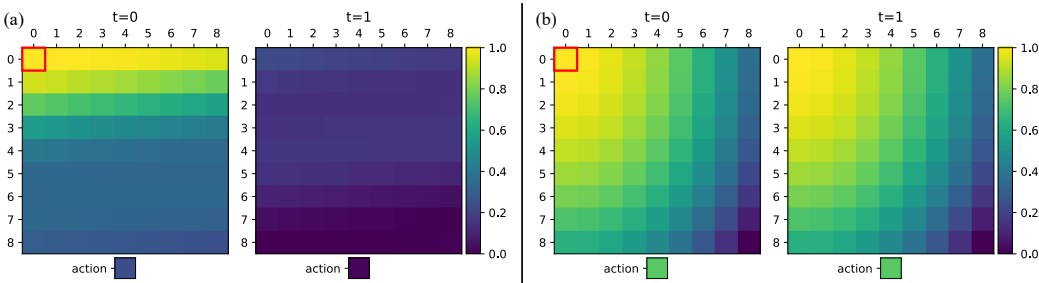

Figure 3: Attention pattern comparison between (a) original RoPE and (b) our relative positional encoding. The heatmaps show a normalized attention score given identical query and key matrices, with respect to the top left query at $(x, y, t) = (0, 0, 0)$, marked with red boxes. Original RoPE tends to attend to $x$-adjacent tokens more than $y$-adjacent and $t$-adjacent tokens. Our spatio-temporal positional encoding preserves adjacency of all three axes.

The codebook is constructed by sampling patches from the replay buffer. A patch is added if it is sufficiently far away from all existing codes: when $\min_{1 \leq i \leq K} \|p - c_i\|_2^2 > \tau$ for a chosen threshold $\tau$. To convert tokens back to images, the tokenizer retrieves the corresponding code for each token $\mathrm{dec}(q) = c_q$ and reassembles the grid into the full image.

## 3 METHOD

Based on the concepts presented in Section 2, our method centers around a transformer world model that exploits spatial relationships within frames and temporal relationships between frames. After the tokenizer converts states and actions to tokens, the token embeddings are augmented with positional encodings that capture spatio-temporal information, before being fed into a transformer. Finally, the transformer output tokens are used by an optimal transport solver to produce the next state tokens, as shown in Figure 1(b). Through this process, the world model generates imagined trajectories for policy training.

### 3.1 SPATIO-TEMPORAL POSITIONAL ENCODING

After converting image frames into tokens (Section 2.4), positional encodings are added to the token embeddings for input to the transformer. Previous transformer world models employ Rotary Position Embedding (RoPE) for positional encoding (Su et al., 2024). However, RoPE uses a single-dimensional position index, which is unable to distinguish between temporal differences (i.e., tokens from different time steps) and spatial differences (i.e., tokens from different positions within the same frame). To incorporate both spatial and temporal information into the model, our method employs a two-fold positional encoding strategy that uses both absolute position and relative position (see Figure 2). First, each token receives a trainable embedding according to its absolute spatial coordinates $(x, y)$ in the grid of patches. This embedding is added to the original token embedding, anchoring each token to a specific semantic location in the observation.

To capture relative positional relationships, our method applies RoPE across spatial and temporal axes. Each token's embedding is divided into three sub-vectors corresponding to its temporal, vertical, and horizontal coordinates. RoPE is then applied independently along each axis, enabling the attention mechanism to capture localized relational structure across both space and time. This formulation allows the model to generalize over local interactions (e.g., neighboring pixels or frames), regardless of absolute location. It preserves adjacency in both spatial and temporal dimensions, while original RoPE loses the adjacency of the $y$-axis and temporal axis, as visualized in Figure 3.

A special indexing strategy handles action tokens, since actions do not have inherent spatial coordinates. To apply relative spatio-temporal encoding, the action at timestep $t$ is assigned the spatial coordinates $(t, t)$. This simplifies action representation to a 1D RoPE formulation, while placing an action token adjacent to the state tokens in the same timestep. Additional indexing details are described in Appendix C.

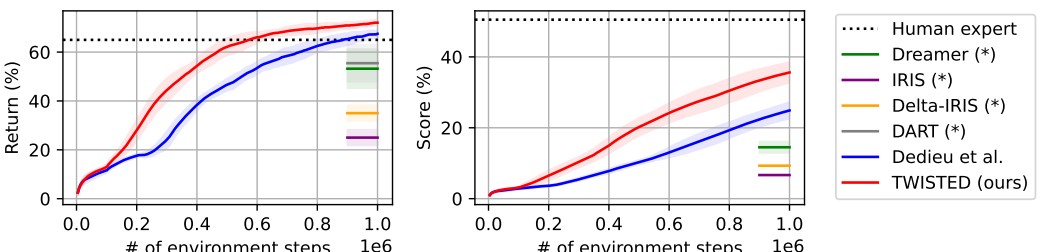

Figure 4: TWISTED achieves state-of-the-art return and score in Craftax-classic, with significantly faster convergence (Matthews et al., 2024). Shading indicates standard deviation among seeds. *Baselines with reported results at 1M steps are displayed with horizontal lines from 900K to 1M steps. DART does not report score, and IRIS and $\Delta$-IRIS do not report standard deviation for score.

## 3.2 OPTIMAL TRANSPORT-BASED DECODING

Having established the spatio-temporal positional encoding for transformer inputs, the next key mechanism involves decoding the transformer outputs. In existing approaches, the output of the transformer world model is directly used to predict each token in the next frame (Micheli et al., 2022; 2024; Agarwal et al., 2024; Dedieu et al., 2025). However, in most visual environments, two adjacent frames are often very similar, e.g. the same tiles but shifted when the player moves right. Our intuition is closely related to the notion of optical flow from classic computer vision tasks (Brox et al., 2004; Vedula et al., 2005; Perazzi et al., 2016). This relation allows tokens to be taken directly from the previous frame into the next frame, rather than offloading the burden of regenerating all next-state tokens to the transformer. To exploit this, the final next-state token predictions are formulated as an optimal transport problem.

Let $L$ be the number of tokens for each frame state. Our method constructs a graph $\mathcal{G} = (\mathcal{V}, \mathcal{E})$, where the vertices $\mathcal{V} = \mathcal{V}_S \cup \mathcal{V}_D$ consist of source vertices $\mathcal{V}_S$ that correspond to previous state tokens and candidate next-state tokens, and destination vertices $\mathcal{V}_D$ that represent the finalized next-state tokens ($|\mathcal{V}_S| = 2L$ and $|\mathcal{V}_D| = L$). The edges $\mathcal{E} = \{(u, v) \mid u \in \mathcal{V}_S, v \in \mathcal{V}_D\}$ connect all sources to all destinations. We now define affinities on these edges for transport.

Let $K$ be the size of the codebook. Given transformer predictions $\mathbf{p}_j \in [0, 1]^K$ for the next state tokens, and previous state tokens $\mathbf{u}_i \in \{0, 1\}^K$ for all $i, j \in \{0, \ldots, L-1\}$, we define an affinity matrix $\boldsymbol{A}^{(prev)} \in \mathbb{R}^{L \times L}$ that scores the affinity between previous state tokens and predicted next-state tokens. Each entry is computed as:

$$A_{ij}^{(prev)} = \langle \mathbf{p}_j, \mathbf{u}_i \rangle - c_d D\left((x_i, y_i), (x_j, y_j)\right), \forall i, j \in \{0, \ldots, L-1\}, \tag{1}$$

where $c_d$ is a coefficient of cost for distance, $D(\cdot)$ is a distance function for 2D coordinates, and $(x_i, y_i)$ and $(x_j, y_j)$ are the 2D coordinates of the $i$-th and $j$-th tokens, respectively. To allow the model to generate new content not present in the previous frame, the graph includes wildcard tokens. The matrix $\boldsymbol{A}^{(gen)} \in \mathbb{R}^{L \times L}$ scores the bonus of admitting newly generated tokens instead of reusing the previous ones, using diagonal entries:

$$A_{kj}^{(gen)} = \begin{cases} \|\mathbf{p}_j\|_\infty - c_w, & \text{if } k = j, \\ -\infty & \text{otherwise}, \end{cases} \quad \forall k, j \in \{0, \ldots, L-1\}, \tag{2}$$

where $c_w$ is a constant penalty for using a wildcard token. With the matrices defined above, an optimal transport plan $\boldsymbol{P}^{(prev)}$ and $\boldsymbol{P}^{(gen)}$ is computed by optimizing the following equation:

$$\begin{aligned}
\underset{\substack{\boldsymbol{P}^{(prev)} \in [0,1]^{L \times L} \\ \boldsymbol{P}^{(gen)} \in [0,1]^{L \times L}}}{\text{minimize}} \quad & \left\langle -\begin{pmatrix} \boldsymbol{A}^{(prev)} \\ \boldsymbol{A}^{(gen)} \end{pmatrix}, \begin{pmatrix} \boldsymbol{P}^{(prev)} \\ \boldsymbol{P}^{(gen)} \end{pmatrix} \right\rangle \\
\text{subject to} \quad & \boldsymbol{P}^{(prev)} \mathbf{1}_L \leq \mathbf{1}_L, \\
& \boldsymbol{P}^{(gen)} \mathbf{1}_L \leq \mathbf{1}_L, \\
& \left( \boldsymbol{P}^{(prev)} + \boldsymbol{P}^{(gen)} \right)^\top \mathbf{1}_L = \mathbf{1}_L.
\end{aligned} \tag{3}$$

Solving the optimal transport problem yields a partial transport plan, represented by a matrix with continuous values in the range $[0,1]$. However, our application requires a strict one-to-one mapping between discrete tokens. To address this, we convert the partial transport plan into a binary assignment matrix with values $\{0,1\}$ using a greedy binarization procedure based on column-wise argmax. Specifically, for each column in the transport matrices $\boldsymbol{P}^{(prev)}$ and $\boldsymbol{P}^{(gen)}$, we identify the row with the highest transport weight, selecting that row in either $\boldsymbol{P}^{(prev)}$ or $\boldsymbol{P}^{(gen)}$, whichever yields the larger value. In the event of a conflict where multiple columns select the same row, we retain the assignment corresponding to the column with the higher transport value and reassign the conflicting column using argmax again, excluding rows that have already been assigned. The complete binarization procedure is described in Algorithm 3 in Appendix D.

Let $\boldsymbol{\Pi}^{(prev)} \in \{0,1\}^{L \times L}$ and $\boldsymbol{\Pi}^{(gen)} \in \{0,1\}^{L \times L}$ denote the binarized versions of $\boldsymbol{P}^{(prev)}$ and $\boldsymbol{P}^{(gen)}$, respectively. The $j$-th token of the next state is determined by copying the $i$-th token of the previous state where $\Pi_{ij}^{(prev)} = 1$. If no such $i$ exists, which occurs only when $\Pi_{jj}^{(gen)} = 1$, the model instead samples from the transformer's predicted distribution. The overall decoding rule is thus defined as

$$\mathbf{u}'_j = \begin{cases} \mathbf{u}_i, & \text{where } \Pi_{ij}^{(prev)} = 1, \\ \text{sample}(\mathbf{p}_j) & \text{where } \Pi_{jj}^{(gen)} = 1, \end{cases} \quad \forall j \in \{0, \dots, L-1\}. \tag{4}$$

Solving this optimization problem involves the Sinkhorn algorithm. By default, the Sinkhorn algorithm minimizes the objective given by a cost matrix rather than an affinity matrix, so the cost matrix is set as the negative of the computed affinity matrix. The end-to-end decoding process is characterized in Algorithm 1 in Appendix D, which also contains additional algorithmic details.

## 4 EXPERIMENTS

### 4.1 CRAFTAX-CLASSIC

**Environment**   We evaluate our method on the Craftax-classic environment (Matthews et al., 2024). Craftax-classic is a fast implementation of Crafter, a challenging procedurally generated, partially observable environment featuring stochastic transitions and a complex hierarchy of achievements (Hafner, 2021). These attributes demand both strong generalization and the ability to model object interactions across time.

**Experiment configuration**   Each method is trained on Craftax-classic for 1M environnment steps, using 10 different seeds per method. The baseline methods consist of DreamerV3 (Hafner et al., 2023), IRIS (Micheli et al., 2022), Δ-IRIS (Micheli et al., 2024), DART (Agarwal et al., 2024), and Dedieu et al. (2025)[1], which had the previous state-of-the-art return on Craftax-classic. Each experiment runs on a single Nvidia RTX 3090 GPU for 57.7 hours. See Appendix B for all hyperparameters and Appendix H for details on compute time.

**Results**   Figure 4 shows that our proposed world model leads to substantially higher return and score, along with faster convergence compared to baseline methods.[2] Return and score are reported

---

[1]We use the (fast) variant from Dedieu et al. (2025), as the (slow) variant is prohibitively expensive to train.

[2]Score is a metric defined as the geometric mean of the success rates for each achievement (Hafner, 2021). Score puts more emphasis on unlocking a variety of achievements, in contrast to return, which is simply the sum of rewards for each episode.

Table 1: Results on Craftax-classic after 0.5M and 1M environment interactions. Return is averaged over episodes of the final 50,000 environment interactions to smooth out variance. The final value for Score is reported directly, as it is already a cumulative metric and does not require additional smoothing. Metrics not reported by baselines are marked as —. † uses hyperparameters of TWISTED.

| | @ 0.5M | | @ 1M | |
| --- | --- | --- | --- | --- |
| Method | Return (%) | Score (%) | Return (%) | Score (%) |
| Human expert | — | — | $65.0 \pm 10.5$ | $50.5 \pm 6.8$ |
| DreamerV3 (Hafner et al., 2023) | — | — | $53.2 \pm 8.0$ | $14.5 \pm 1.6$ |
| IRIS (Micheli et al., 2022) | — | — | $25.0 \pm 3.2$ | 6.66 |
| $\Delta$-IRIS (Micheli et al., 2024) | — | — | $35.0 \pm 3.2$ | 9.30 |
| DART (Agarwal et al., 2024) | — | — | $55.45 \pm 3.39$ | — |
| Dedieu et al. (2025) | — | — | $67.42 \pm 0.55$ | $27.91 \pm 0.63$ |
| Dedieu et al. (2025) (reproduced) | $48.17 \pm 0.82$ | $10.22 \pm 0.20$ | $68.14 \pm 0.42$ | $24.89 \pm 0.74$ |
| Dedieu et al. (2025) (reproduced)† | $54.32 \pm 0.60$ | $13.06 \pm 0.39$ | $68.55 \pm 0.72$ | $27.24 \pm 0.86$ |
| TWISTED (ours) | $\mathbf{63.10} \pm 1.24$ | $\mathbf{20.12} \pm 0.80$ | $\mathbf{72.46} \pm 0.45$ | $\mathbf{35.60} \pm 0.92$ |

in Table 1, as the mean and standard error over 10 seeds. After 1M environment interactions, our method achieves a final return and score surpassing all baselines. It also outperforms the previous best baseline during training at 0.5M environment interactions, demonstrating superior sample efficiency in a more data-constrained setting.

**Ablations** To further understand our method's performance, we conduct ablation studies to evaluate the individual contributions of the spatio-temporal positional encoding (STPE) and the optimal transport mechanism. Table 2 shows that both components make independent improvements to policy return and score, but using them together leads to the best result. Furthermore, for spatio-temporal positional encoding, including absolute spatial embeddings improves the performance compared to only using relative encoding (Relative PE only).

Table 2: Ablations on Craftax-classic with 1M interactions. † uses hyperparameters of TWISTED.

| Method | Return (%) | Score (%) |
| --- | --- | --- |
| Dedieu et al. (2025)† | $68.55 \pm 0.72$ | $27.24 \pm 0.86$ |
| Relative PE only | $69.26 \pm 0.50$ | $29.37 \pm 0.80$ |
| STPE only | $71.85 \pm 0.63$ | $33.94 \pm 1.10$ |
| Optimal transport only | $69.77 \pm 0.65$ | $31.08 \pm 0.88$ |
| TWISTED (ours) | $\mathbf{72.46} \pm 0.45$ | $\mathbf{35.60} \pm 0.92$ |

**Accuracy evaluation** To assess the contribution of optimal transport to world model prediction, we measure the prediction accuracy with and without the transport mechanism. Our evaluation uses 10,000 environment transitions and counts how many next states are predicted perfectly (where every predicted token is correct). Table 3 shows that adding optimal transport improves accuracy. The transformer alone has particularly low accuracy in cases involving randomly moving creatures, which optimal transport helps with. By improving the accuracy of world model prediction, optimal transport decoding leads to higher quality imaginations and improved policy performance as seen in the ablations.

**Qualitative analysis** Figure 5 compares imaginations generated by our method vs. Dedieu et al. (2025). Our method excels in situations where tiles in the generated frame are correlated. For example, a creature in Craftax-classic can move to adjacent tiles, but it should only move to one destination tile and should not be duplicated to multiple destination tiles. However, because the transformer generates output tokens for a state in parallel, it cannot capture this constraint naturally. Therefore, during imagination, duplication or disappearance of creatures occurs, which is a critical defect of modeling environment dynamics. Optimal transport-based decoding eliminates this issue by capturing the appropriate constraint between output tiles. Solving this issue is particularly important because similar hallucinations arise in non-transformer world models as well (see Appendix A).

Table 3: Prediction accuracy on a dataset of 10,000 transitions. The first column reports overall accuracy, while the latter two break down accuracy based on whether the input state contains a randomly moving creature. Applying the optimal transport mechanism to the STPE-only transformer outputs increases accuracy by 3.39%. The transformer accuracy is especially low for transitions involving creatures, which optimal transport improves by 3.45%. † uses hyperparameters of TWISTED.

| Method | Accuracy (%) | w/ creatures (%) | w/o creatures (%) |
|---|---|---|---|
| Dedieu et al. (2025)$^{\dagger}$ | 46.94 | 33.83 | 61.03 |
| STPE only | 47.74 | 34.92 | 62.13 |
| TWISTED (ours) | **51.13** | **38.37** | **65.46** |

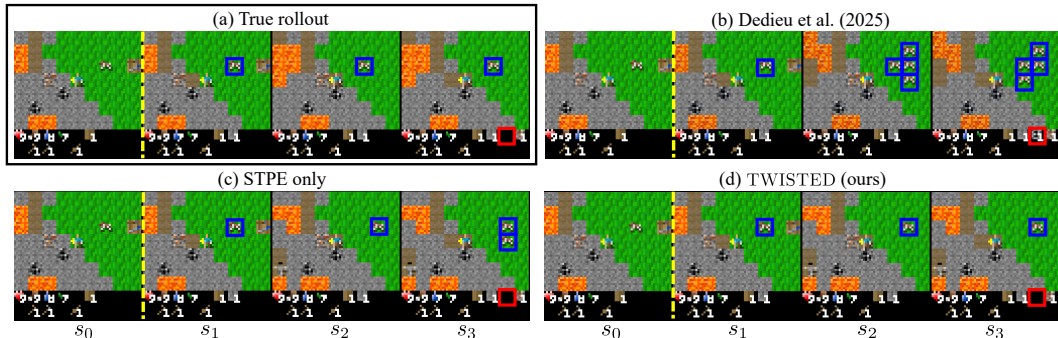

Figure 5: Comparison of imagined rollouts from different world models. (a) shows the ground-truth environment trajectory, while (b), (c), and (d) illustrate imagined rollouts generated by the baseline, the baseline with spatio-temporal positional encoding (STPE only), and TWISTED, respectively. All rollouts begin from the same initial state $s_0$ (left of the yellow dashed line). TWISTED fixes inaccurate dynamics (red boxes) and duplication errors (blue boxes) produced by the baseline.

## 4.2 CRAFTAX

We also evaluate our method on Craftax, a more complex and difficult environment that builds on Craftax-classic (Matthews et al., 2024). Craftax features a larger screen, more items, more enemies, and more levels compared to Craftax-classic (details in Appendix E). We compare against Simulus (Cohen et al., 2025) and Dedieu et al. (2025)[3], which set the previous best return and score, respectively. Table 4 reports return and score on Craftax, as the mean and standard error over 5 seeds. TWISTED

Table 4: Results on Craftax after 1M environment interactions. Simulus does not report Score (—).

| Method | Return (%) | Score (%) |
|---|---|---|
| Dedieu et al. (2025) | $5.44 \pm 0.25$ | $1.53 \pm 0.10$ |
| Simulus | 6.59 | — |
| TWISTED (ours) | $\mathbf{7.09 \pm 0.20}$ | $\mathbf{2.40 \pm 0.04}$ |

achieves a return of 7.09% and a score of 2.40%, surpassing the baselines. These results demonstrate that TWISTED can generalize to more difficult environments.

## 4.3 MINATAR

To further validate the generalization performance of our approach, we also evaluate on the MinAtar benchmark (Young & Tian, 2019; Lange, 2022). MinAtar consists of 4 Atari games with simplified symbolic observations of size $10 \times 10$. We compare against the previous state of the art for model-based RL, Dedieu et al. (2025), and the recent model-free Artificial Dopamine (AD) agent (Guan et al., 2023). Each method is trained on each game in MinAtar for 1M environment steps (except AD uses 5M steps), using 10 seeds per game. Table 5 shows that TWISTED outperforms the

---

[3]We report the (fast) variant from version arXiv:2502.01591v1 of Dedieu et al. (2025).

Table 5: Returns on MinAtar after 1M environment interactions (or 5M for AD). Return is evaluated on 1,000 evaluation episodes at the end of training.

| Method | Asterix | Breakout | Freeway | SpaceInvaders |
|---|---|---|---|---|
| AD (Guan et al., 2023) | $21.05 \pm 0.65$ | $27.78 \pm 0.16$ | $57.68 \pm 0.07$ | $140.36 \pm 1.70$ |
| Dedieu et al. (2025) | $44.81 \pm 3.54$ | $93.92 \pm 1.44$ | $71.12 \pm 0.13$ | $186.16 \pm 1.25$ |
| TWISTED (ours) | $\mathbf{50.04} \pm 2.98$ | $\mathbf{99.53} \pm 2.31$ | $\mathbf{71.34} \pm 0.07$ | $\mathbf{188.85} \pm 0.62$ |

Table 6: Aggregate metrics on Atari 100K after 100K environment interactions. Return for each game is evaluated on 100 evaluation episodes at the end of training.

| Method | IQM ($\uparrow$) | Optimality Gap ($\downarrow$) | Mean ($\uparrow$) | Median ($\uparrow$) |
|---|---|---|---|---|
| DreamerV3 (Hafner et al., 2023) | 0.487 | 0.510 | 1.124 | 0.485 |
| STORM (Zhang et al., 2023) | 0.561 | 0.472 | 1.222 | 0.425 |
| Diamond (Alonso et al., 2024) | 0.641 | 0.480 | 1.459 | 0.373 |
| Simulus (Cohen et al., 2025) (reproduced) | 0.969 | 0.410 | **1.636** | 0.739 |
| TWISTED (ours) | **1.092** | **0.376** | 1.616 | **0.978** |

baselines in all 4 games. Return graphs for each game can be found in Appendix F. By improving in every game, TWISTED demonstrates that spatio-temporal encoding and optimal transport-based decoding confer robust benefits across a variety of environments.

## 4.4 ATARI 100K

We also evaluate on the popular Atari 100K benchmark, which trains on a suite of 26 Atari games for 100K environment interactions each, equivalent to 2 hours of human gameplay (Kaiser et al., 2020). We use the current state-of-the-art token-based world model, Simulus (Cohen et al., 2025), as our baseline, since the Dedieu et al. (2025) baseline is not designed for or tested on Atari 100K. We create an instantiation of TWISTED for Atari 100K by applying spatio-temporal positional encoding and optimal transport-based decoding to Simulus. Each method is trained on each game in Atari 100K for 100K environment steps, using 5 seeds per game. Results on Atari 100K are reported as human-normalized score, calculated as $\frac{\text{agent\_return}-\text{random\_agent\_return}}{\text{human\_return}-\text{random\_agent\_return}}$. Table 6 shows that TWISTED exceeds the baseline and achieves new state-of-the-art performance in interquartile mean (IQM), optimality gap, and median. Detailed results for each game are presented in Table 15 of Appendix I. By excelling in Atari 100K, TWISTED shows that its performance generalizes across 2D visual RL environments.

## 5 RELATED WORKS

**Transformer world models** Transformer architectures have been effectively utilized in model-based RL. The concept of transformer world models was first introduced by IRIS (Micheli et al., 2022). Building upon IRIS, $\Delta$-IRIS proposed an agent architecture that encodes stochastic deltas between time steps, enhancing token efficiency by exploiting similarities between adjacent frames (Micheli et al., 2024). TWM, STORM, DART, and TWISTER also utilized the transformer architecture for world models, demonstrating its efficacy across different benchmarks (Robine et al., 2023; Zhang et al., 2023; Agarwal et al., 2024; Burchi & Timofte, 2025). Transformer world models further advanced with techniques including nearest neighbor tokenization and block teacher forcing, achieving state-of-the-art performance on Craftax-classic (Dedieu et al., 2025). Outside of transformers, other world models have used GRUs (DreamerV3), diffusion (DIAMOND), decoder-free latent spaces (TD-MPC2), and discrete codebook latent spaces (DC-MPC) (Hafner et al., 2023; Alonso et al., 2024; Hansen et al., 2024; Scannell et al., 2025).

**Positional embeddings in video modeling**  Positional encoding for multi-dimensional information has been developed in the context of video modeling. DFoT, CogVideoX, and HunyuanVideo adopted a 3D-RoPE formulation in which 1D-RoPE is applied independently along each axis, and the resulting axis-wise encodings are concatenated to form the final positional representation (Song et al., 2025; Yang et al., 2025; Kong et al., 2024). The Qwen2-VL series introduced Multimodal Rotary Position Embedding (M-RoPE), decomposing positional embeddings into components capturing 1D textual and 3D video information (Wang et al., 2024). Complementing this, VideoRoPE proposed enhancements such as Low-frequency Temporal Allocation and Adjustable Temporal Spacing to improve video rotary position embeddings, demonstrating superior performance in video understanding tasks (Wei et al., 2025). Cosmos proposed an world model with 3D-factorized RoPE, generating video using spatial and temporal information (Agarwal et al., 2025). However, these video-based 3D RoPE methods cannot be directly applied in an RL context because they do not support spatio-temporal embeddings for actions.

**Optimal transport in RL**  Optimal transport theory has been applied to RL in other contexts, specifically for curriculum and offline reinforcement learning. CurrOT framed curriculum generation as a constrained optimal transport problem between task distributions (Klink et al., 2022). GRADIENT formulated curriculum reinforcement learning as an optimal transport problem with a tailored distance metric between tasks (Huang et al., 2022). Additionally, Achievement Distillation introduced a contrastive learning method using optimal transport to enhance the discovery of hierarchical achievements, leading to improved sample efficiency (Moon et al., 2023).

## 6 CONCLUSION

In this paper, we present TWISTED, a transformer world model tailored for 2D visual RL environments. TWISTED captures the inherent structure of visual environment inputs by encoding both spatial and temporal dimensions using a combination of absolute and relative positional encodings. By selectively reusing tokens from preceding frames with optimal transport-based decoding, it effectively leverages frame-to-frame similarities to model next-state tokens instead of solely relying on the transformer to regenerate each one. These innovations enable TWISTED to achieve new state-of-the-art performance on the challenging Craftax-classic, Craftax, MinAtar, and Atari 100K benchmarks.

REPRODUCIBILITY STATEMENT

For full reproducibility, all source code is included in the supplementary materials. The source code will also be released on GitHub upon acceptance. All implementation details are described in Appendix B for the world model and policy, Appendix C for spatio-temporal encoding, and Appendix D for optimal transport. All hyperparameters are listed in Appendix B for Craftax-Classic, Appendix E for Craftax, and Appendix F for MinAtar.

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

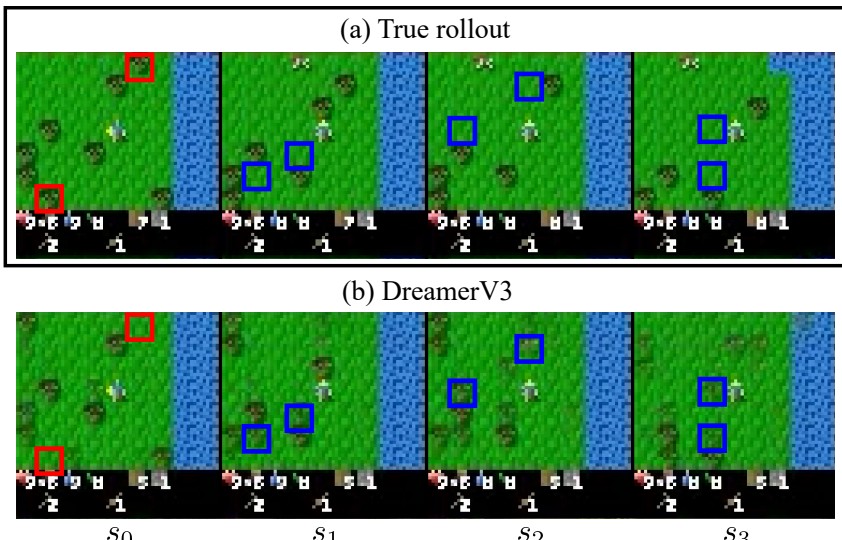

Figure 6: An imagination rollout of DreamerV3 compared to the ground-truth trajectory. DreamerV3's imagination includes disappearance of trees (red boxes) and duplication of trees (blue boxes) over time, similar to duplication issues shown in Figure 5 for Dedieu et al. (2025)

## A  ADDITIONAL QUALITATIVE ANALYSIS

The qualitative analysis in Section 4.1 discusses how parallel decoding in baseline transformer world models leads to duplication and disappearance of objects during imagination. However, this issue is not isolated to transformers and can also be observed in non-transformer world models, like DreamerV3 (Hafner et al., 2023). Figure 6 shows duplication and disappearance artifacts in an imagination rollout generated by DreamerV3, demonstrating that this issue occurs across different types of world models. Thus, by eliminating these hallucinations, optimal transport-based decoding resolves a problem that is widespread among world models.

## B  AGENT TRAINING AND IMPLEMENTATION

### B.1  TRAINING LOOP

This section outlines the training procedure for the world model and the policy, which are trained concurrently through alternating update steps. The overall training loop is composed of the following steps:

1. **Environment interaction:** Execute the current policy in the real environment and store the resulting experiences in a replay buffer.

2. **Policy update on real data:** Update the policy using the most recent real environment experiences collected in Step 1. The policy is trained on the data over $E_{\text{env}}$ epochs, with each batch split into $B_{\text{policy}}$ minibatches due to memory constraints.

3. **Tokenizer training:** Sample experiences from the replay buffer to train the nearest neighbor tokenizer. The tokenizer is updated on $U_{\text{tokenizer}}$ batches of sample trajectories.

4. **World model training:** Sample experiences from the replay buffer to train the transformer world model. The world model is updated on $U_{\text{WM}}$ batches of sample trajectories, using $B_{\text{WM}}$ minibatches.

5. **Policy update in imagination:** For training steps $t > T_{\text{warmup}}$, generate $U_{\text{imag}}$ batches of imagined trajectories using the world model and the current policy, and update the policy on these synthetic rollouts. During the initial $T_{\text{warmup}}$ real environment interactions, this step is skipped to allow the world model to reach sufficient accuracy before generating

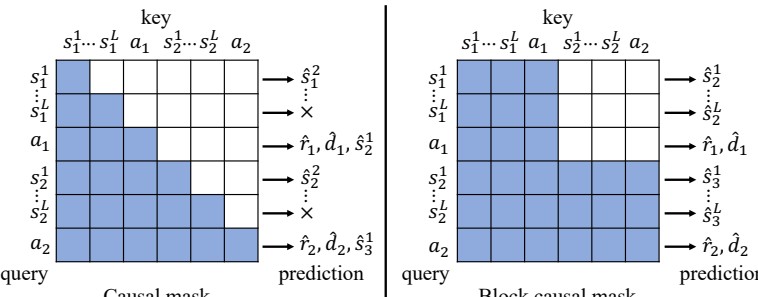

Figure 7: Comparison between the causal attention mask and the block causal attention mask. The token $s_t^i$ denotes the $i$-th state token at timestep $t$, $a_t$ denotes the action, $\hat{r}_t$ denotes the predicted reward, and $\hat{d}_t$ denotes the predicted done signal. Only two state tokens are shown per state for simplicity. (left) In the causal mask, each token attends to the tokens preceding it. The output embeddings of state token $s_t^i$ are used to predict the subsequent state token $s_t^{i+1}$. The reward $\hat{r}_t$ and done signal $\hat{d}_t$ are predicted from $a_t$, and the output of $s_t^L$ is unused. (right) In the block causal mask, all state and action tokens in the same timestep attend to each other, and they are used to predict the corresponding token in the next timestep ($a_t$ predicts $\hat{r}_t$ and $\hat{d}_t$). This allows each frame to be predicted in parallel rather than token-by-token.

imaginations. At the start of each imagination rollout, the policy uses $T_{\text{burn}}$ frames from the replay buffer to initialize its RNN hidden state.

The overall training loop is repeated until the agent has performed a total of $T_{\text{total}}$ real environment interactions.

### B.2 WORLD MODEL ARCHITECTURE AND LOSS

Our transformer world model follows the GPT-2 architecture (Radford et al., 2019). The model operates over tokenized sequences that encode states and actions over $T$ consecutive frames. These tokens are first mapped to 128-dimensional embeddings via a learned embedding layer. Absolute positional embeddings are then added, followed by an initial dropout layer. The resulting embeddings are processed through a stack of three transformer blocks. Each block consists of the following components:

1. Layer normalization
2. Multi-head attention module, comprising:
   (a) Self-attention with a block causal mask. In the block causal mask, tokens within the same timestep are decoded in parallel (see Figure 7) (Dedieu et al., 2025).
   (b) A linear projection to the 128-dimensional embedding space
   (c) Dropout
3. Residual connection with the block input
4. Layer normalization
5. Feed-forward multilayer perceptron (MLP) composed of:
   (a) A hidden layer of dimension 512
   (b) GeLU activation
   (c) Dropout

After processing through the final block, the output undergoes a final layer normalization and is then passed to three separate prediction heads: one for the next state tokens, one for the reward signal, and one for the done signal. We denote the output embeddings as

$$(E_1^1, \ldots, E_1^{L+1}, E_2^1, \ldots, E_2^{L+1}, \ldots, E_T^1, \ldots, E_T^{L+1}).$$

where $L$ represents the number of state tokens per frame, and $E_t^i$ corresponds to the $i$-th output embedding at timestep $t$. These embeddings are routed to prediction heads as follows:

1. For $i \leq L$, the embedding $E_t^i$ is input to the observation head, an MLP comprising a 128-dimensional linear layer, a ReLU activation, and a final linear layer projecting to the codebook size $K$. The output logits define a categorical distribution over the $K$ possible values of the predicted state token $s_{t+1}^i$.

2. The embedding $E_t^{L+1}$, corresponding to the position of the action token, is passed to both the reward and done heads. Each head is an MLP consisting of a 128-dimensional linear layer, a ReLU activation, and a final linear layer projecting to two output classes. Although the Craftax-classic environment defines reward values of $-0.1$, $0.1$, and $1.0$, we follow Dedieu et al. (2025) and binarize the reward signal to improve stability, ignoring the $-0.1$ and $0.1$ cases.

The model is trained on trajectories of length $T_{\text{WM}}$ sampled from the replay buffer. The total loss is the sum of three components:

1. Cross-entropy loss over next-state token predictions (across $K$ classes).

2. Cross-entropy loss for binary reward classification (0 or 1).

3. Cross-entropy loss for done signal prediction.

Optimization is performed using the Adam optimizer with gradient norm clipping to stabilize training (Kingma & Ba, 2015). Hyperparameters for architecture and training are provided in Table 7.

Table 7: World model hyperparameters. Sweep range indicates the values tried per hyperparameter, with the final Value being chosen based on highest return.

| Area | Hyperparameter | Value | Sweep range |
|---|---|---|---|
| Architecture | Sequence length $T_{\text{WM}}$ | 20 | |
| | State tokens per frame $L$ | 81 | |
| | Number of blocks | 3 | |
| | Number of attention heads | 8 | |
| | Embedding dimension | 128 | |
| | MLP hidden layer dimension | 512 | |
| | Dropout rate | 0.1 | |
| | Attention mask | Block causal | |
| | Inference with key-value caching | True | |
| Optimal transport | Distance cost coefficient $c_d$ | 0.6 | {0.0, 0.3, 0.6, 0.8} |
| | Wildcard cost $c_w$ | 0.3 | {0.3, 0.6} |
| | Sinkhorn regularization parameter $\epsilon$ | 0.00001 | {0.00001, 0.0001} |
| | Sinkhorn iterations | 10 | {10, 100, 500} |
| Training | Number of updates $U_{\text{WM}}$ | 500 | |
| | Number of minibatches $B_{\text{WM}}$ | 3 | |
| | Replay buffer size | 128,000 | |
| Optimization | Optimizer | Adam | |
| | Learning rate | 0.001 | |
| | Max norm for gradient clipping | 0.5 | |
| Tokenizer | Codebook size $K$ | 4096 | |
| | Single patch shape | $7 \times 7 \times 3$ | |
| | New code threshold $\tau$ | 0.75 | |
| | Number of updates $U_{\text{tokenizer}}$ | 25 | |

Table 8: Policy hyperparameters. Sweep range indicates the values tried per hyperparameter, with the final Value being chosen based on highest return.

| Area | Hyperparameter | Value | Sweep range |
|---|---|---|---|
| Environment | Environment interactions $T_{\text{total}}$ | 1,000,000 | |
| | Warmup interactions $T_{\text{warmup}}$ | 50,000 | {50k, 100k, 200k} |
| | Number of environments (batch size) | 48 | |
| | Rollout horizon in environment | 96 | |
| | Rollout horizon in imagination $T_{\text{WM}}$ | 20 | |
| | Burn-in horizon for RNN in imagination $T_{\text{burn}}$ | 5 | |
| Training | Number of updates in imagination $U_{\text{imag}}$ | 300 | {150, 300, 600, 1200} |
| | Number of epochs in environment $E_{\text{env}}$ | 4 | |
| | Number of epochs in imagination | 1 | |
| | Number of minibatches in environment $B_{\text{policy}}$ | 8 | |
| | Number of minibatches in imagination | 1 | |
| PPO | Discount factor $\gamma$ | 0.925 | |
| | TD weight $\lambda$ | 0.625 | |
| | Clipping value $\epsilon$ | 0.2 | |
| | TD loss coefficient $\lambda_{\text{TD}}$ | 2.0 | |
| | Entropy loss coefficient $\lambda_{\text{ent}}$ | 0.01 | |
| | PPO target discount factor $\alpha$ | 0.95 | |
| Optimization | Optimizer | Adam | |
| | Learning rate | 0.00045 | |
| | Max norm for gradient clipping | 0.5 | |

## B.3 POLICY NETWORK ARCHITECTURE AND LOSS

### B.3.1 ARCHITECTURE

We adopt the policy network architecture introduced in Dedieu et al. (2025), which comprises three primary components: a convolutional encoder, a recurrent neural network (RNN), and separate MLP heads for action and value prediction.

The convolutional encoder consists of three convolutional blocks with channel sizes [64, 64, 128]. Each block contains an instance normalization layer, a $3 \times 3$ convolutional layer with stride 1, a $3 \times 3$ max-pooling layer with stride 2, and two ResNet-style sub-blocks. Each ResNet block includes a ReLU activation, instance normalization, a 3×3 convolution with stride 1, and a skip connection to preserve the input. The encoder produces an output of shape $8 \times 8 \times 128$, which is flattened into a 8192-dimensional vector, denoted by $z$. The vector $z$ is then projected into a 256-dimensional representation through a ReLU activation, a linear layer, and layer normalization. This projected representation serves as input to a GRU recurrent module, which outputs a vector $y \in \mathbb{R}^{256}$ along with the updated hidden state $h \in \mathbb{R}^{256}$.

The action and value heads share an identical structure except for the final output projection. Each head takes the concatenated vector $[z, y]$ as input and applies a sequence of transformations: ReLU activation, layer normalization, a linear projection to 2048, another ReLU activation, and a residual block composed of two linear layers with ReLU activations. The output is passed through a final layer normalization, followed by the task-specific output projection—either to action logits or a scalar value estimate.

### B.3.2 TRAINING

We follow the policy training procedure described in Dedieu et al. (2025), using Proximal Policy Optimization (PPO) (Schulman et al., 2017) as the underlying policy gradient algorithm.

Let the trajectory be denoted as $\tau = (o_{1:T+1}, a_{1:T}, r_{1:T}, d_{1:T}, h_{0:T})$, where $o_t$ represents the observations, $a_t$ the actions, $r_t$ the rewards, $d_t$ the done signals, and $h_t$ the hidden states of the RNN. At each timestep, PPO computes the value estimates $v_{1:T+1} = V_{\Phi_{\text{old}}}(o_{1:T+1})$ and the action probabili-

ties $\pi_{\Phi_{\text{old}}}(a_t|o_t)$ under the current fixed parameters $\Phi_{\text{old}}$. The policy is optimized by minimizing the following PPO objective:

$$\mathcal{L}_{\text{PPO}}(\Phi) = \frac{1}{T}\sum_{t=1}^{T}\Big\{ -\min\left(p_t(\Phi)A_t, \text{clip}(p_t(\Phi))A_t\right)$$
$$+ \lambda_{\text{TD}}(V_\Phi(o_t) - q_t)^2$$
$$- \lambda_{\text{ent}}\mathcal{H}\left(\pi_\Phi(.|o_t)\right)\Big\}$$

where $p_t(\Phi)$ is the probability ratio $\frac{\pi_\Phi(a_t|o_t)}{\pi_{\Phi_{\text{old}}}(a_t|o_t)}$ and $\text{clip}(x)$ is the clipping function $\min(\max(x, 1 - \epsilon), 1 + \epsilon)$. Here, $A_t$ denotes a generalized advantage estimation, $q_t$ is a temporal difference (TD) target, and $\mathcal{H}$ is the entropy operator. The advantages $A_t$ and targets $q_t$ are computed as

$$A_t = \delta_t + (1 - \text{done}_t)\gamma\lambda A_{t+1},$$
$$q_t = A_t + v_t,$$

where $\delta_t = r_t + (1 - \text{done}_t)\gamma v_{t+1} - v_t$.

We incorporate two modifications to the standard PPO implementation:

- Generalized advantage estimates $A_t$ are standardized across training batches to stabilize learning.
- We track the moving average of the mean and standard deviation of $q_t$, with discount factor $\alpha$, and train the value function to predict the standardized targets.

All policy training hyperparameters are shown in Table 8.

## C  RELATIVE SPATIO-TEMPORAL ENCODING IMPLEMENTATION

Our implementation of relative spatio-temporal encoding extends Rotary Position Embedding (RoPE) by adapting it to both spatial and temporal contexts (Su et al., 2024). While RoPE rotates pairs of embedding dimensions using frequencies based on a 1D position index, our method modulates the rotation amount based on three indices, two spatial and one temporal. Our method divides dimension pairs in a 3:1 ratio between spatial and temporal encoding, following the design principle of VideoRoPE (Wei et al., 2025). Pairs associated with lower rotation frequencies are used for temporal encoding and are rotated based on the temporal index. In contrast, pairs with higher rotation frequencies are used for spatial encoding. Given the 2D nature of spatial positions, spatial pairs are further split evenly between the horizontal and vertical axes. These are interleaved across the embedding dimension to ensure balanced representation. As a result, the axes contributing to rotation follow the pattern $(x, y, x, y, \ldots, x, y, t, t, \ldots, t)$, ordered by decreasing rotation frequency. The relative spatio-temporal encoding is implemented by applying block-diagonal rotation matrices to the query and key vectors. The matrix $\mathbf{R}_{xy}$ applies higher-frequency rotations parameterized by the spatial coordinates $(x, y)$, while $\mathbf{R}_t$ applies lower-frequency rotations parameterized by the temporal index $t$. These rotations follow the standard rotary position embedding (RoPE) formulation extended to two spatial dimensions and one temporal dimension.

$$\mathbf{R}_{xy} = \begin{pmatrix} \cos\theta_0 x & -\sin\theta_0 x & 0 & 0 & \cdots & 0 & 0 & 0 & 0 \\ \sin\theta_0 x & \cos\theta_0 x & 0 & 0 & \cdots & 0 & 0 & 0 & 0 \\ 0 & 0 & \cos\theta_1 y & -\sin\theta_1 y & \cdots & 0 & 0 & 0 & 0 \\ 0 & 0 & \sin\theta_1 y & \cos\theta_1 y & \cdots & 0 & 0 & 0 & 0 \\ \vdots & \vdots & \vdots & \vdots & \ddots & \vdots & \vdots & \vdots & \vdots \\ 0 & 0 & 0 & 0 & \cdots & \cos\theta_{k-2} x & -\sin\theta_{k-2} x & 0 & 0 \\ 0 & 0 & 0 & 0 & \cdots & \sin\theta_{k-2} x & \cos\theta_{k-2} x & 0 & 0 \\ 0 & 0 & 0 & 0 & \cdots & 0 & 0 & \cos\theta_{k-1} y & -\sin\theta_{k-1} y \\ 0 & 0 & 0 & 0 & \cdots & 0 & 0 & \sin\theta_{k-1} y & \cos\theta_{k-1} y \end{pmatrix}$$

$$\mathbf{R}_t = \begin{pmatrix} \cos\theta_k t & -\sin\theta_k t & \cdots & 0 & 0 \\ \sin\theta_k t & \cos\theta_k t & \cdots & 0 & 0 \\ \vdots & \vdots & \ddots & \vdots & \vdots \\ 0 & 0 & \cdots & \cos\theta_{D/2-1} t & -\sin\theta_{D/2-1} t \\ 0 & 0 & \cdots & \sin\theta_{D/2-1} t & \cos\theta_{D/2-1} t \end{pmatrix}$$

Given a query vector $\mathbf{q}_i$ for token $i$ and a key vector $\mathbf{k}_j$ for token $j$, their spatio-temporal rotary embeddings are obtained by applying the corresponding spatial and temporal rotations. Let $x(i)$, $y(i)$, and $t(i)$ denote the spatial–temporal coordinates of the $i$-th token. Then the transformed query and key vectors are:

$$\mathbf{q}_i' = \begin{pmatrix} \boldsymbol{R}_{x(i)y(i)} & \mathbf{0} \\ \mathbf{0} & \boldsymbol{R}_{t(i)} \end{pmatrix} \mathbf{q}_i$$

$$\mathbf{k}_j' = \begin{pmatrix} \boldsymbol{R}_{x(j)y(j)} & \mathbf{0} \\ \mathbf{0} & \boldsymbol{R}_{t(j)} \end{pmatrix} \mathbf{k}_j.$$

$$\mathbf{q}_i'^\top \mathbf{k}_j' = \mathbf{q}_i^\top \begin{pmatrix} \boldsymbol{R}_{x(j)-x(i),y(j)-y(i)} & \mathbf{0} \\ \mathbf{0} & \boldsymbol{R}_{t(j)-t(i)} \end{pmatrix} \mathbf{k}_j$$

As action tokens lack inherent spatial coordinates, assigning them fixed spatial positions would limit the effectiveness of relative positional encoding across the majority of embedding dimensions. To address this, spatial coordinates for action tokens are defined along the diagonal, $(t, t)$, where $t$ represents the temporal index. State tokens are assigned spatial coordinates offset from this diagonal, $(x + t, y + t)$, ensuring temporal alignment with action tokens while preserving spatial variation.

To avoid positional collisions between state and action tokens, they are given different temporal indices. That is, the state and action tokens $s_t^1, \ldots s_t^L, a_t, s_{t+1}^1, \ldots, s_{t+1}^L, a_{t+1}$ are given temporal indices $2t, \ldots, 2t, 2t + 1, 2(t + 1), \ldots 2(t + 1), 2(t + 1) + 1$. This staggered assignment ensures that each token occupies a unique spatio-temporal location, maintaining positional distinctiveness throughout the sequence.

## D    OPTIMAL TRANSPORT IMPLEMENTATION

This section details various components of our optimal transport implementation. Algorithm 1 characterizes the end-to-end decoding process. Algorithm 2 describes the Sinkhorn algorithm (Cuturi, 2013). Algorithm 3 describes the process of converting the transport plan's continuous values into binary values. It is adapted from Kim et al. (2020). We use the OTT-JAX library for our Sinkhorn solver implementation (Cuturi et al., 2022).

**Distance cost**    For the affinity matrix $\boldsymbol{A}^{(prev)}$ in Equation 1, we include a distance penalty $D((x_i, y_i), (x_j, y_j))$. This choice of distance cost can be adapted based on the environment. For our environments, we impose a movement constraint by defining the distance cost as

$$D((x_i, y_i), (x_j, y_j)) = \begin{cases} d, & \text{if } d \leq 4 \\ +\infty & \text{otherwise,} \end{cases}$$

$$\text{where } d = \|(x_i, y_i) - (x_j, y_j)\|_2^2.$$

This constraint reflects the fact that in environments like Craftax-classic, Craftax, and MinAtar, a token's spatial displacement between consecutive frames is limited to a maximum of two positions in any direction. For Craftax-classic and Craftax, this accounts for the potential movement of one by the player and one by a creature token, if applicable.

**Choosing between transformer and optimal transport output**    Optimal transport provides an effective mechanism for reusing tokens from the previous frame. However, it is less effective in scenarios where novel tokens must be introduced, such as when the agent moves to a previously unexplored area. In such cases, optimal transport may fail to consistently route wildcard entries to the appropriate newly generated tokens. Conversely, the transformer world model is capable of freely generating new tokens as needed, but lacks a mechanism for directly reusing tokens from prior frames. Rather than committing to a single output modality, we adopt a hybrid strategy for Craftax-classic and Craftax that selects between optimal transport and transformer outputs based on spatial position. In Craftax-classic and Craftax, new visual content appears along the screen boundaries as

---

**Algorithm 1** Decoding with optimal transport

---

**Input:** transformer prediction $\mathbf{p}$,
    previous tokens $\mathbf{u}$,
    number of tokens per frame $L$,
    Sinkhorn regularization parameter $\epsilon$,
    Number of Sinkhorn iterations $T$
**Output:** Generated tokens for next frame $\mathbf{u}'$

Compute $\boldsymbol{A}^{(prev)}, \boldsymbol{A}^{(gen)}$ from Equations 1 and 2

$\boldsymbol{A} = \text{concatenate} \begin{pmatrix} \boldsymbol{A}^{(prev)} & \mathbf{0} \in \mathbb{R}^{L \times L} \\ \boldsymbol{A}^{(gen)} & \mathbf{0} \in \mathbb{R}^{L \times L} \end{pmatrix} \in \mathbb{R}^{(2L) \times (2L)}$      ▷ Concatenate for Sinkhorn input

$\boldsymbol{P} = \text{SINKHORN}(-\boldsymbol{A}, \epsilon, T)$
$\boldsymbol{P}^{(prev)} = \boldsymbol{P}_{1:L,1:L}$
$\boldsymbol{P}^{(gen)} = \boldsymbol{P}_{L+1:2L,1:L}$

$\boldsymbol{\Pi}^{(prev)}, \boldsymbol{\Pi}^{(gen)} = \text{BINARIZATION}(\boldsymbol{P}^{(prev)}, \boldsymbol{P}^{(gen)})$

**for** $j = 0$ **to** $L - 1$ **do**         ▷ Finalize the state prediction
    **if** $\Pi_{ij}^{(prev)} = 1$ for some $i$ **then**
        $\mathbf{u}'_j = \mathbf{u}_i$         ▷ Take a token from the previous state
    **else if** $\Pi_{jj}^{(gen)} = 1$ **then**
        $\mathbf{u}'_j = \text{sample}(\mathbf{p}_j)$     ▷ Use a generated token from the transformer
    **end if**
**end for**
**return** $\mathbf{u}'$

---

**Algorithm 2** SINKHORN implementation

---

**Input:** Cost matrix $\boldsymbol{C}$,
    Sinkhorn regularization parameter $\epsilon$,
    Number of Sinkhorn iterations $T$
**Output:** Optimal transport plan $\boldsymbol{P}$

$\boldsymbol{K} = \exp(\boldsymbol{C}/\epsilon)$
Set uniform marginals: $\mathbf{r} = \frac{1}{\text{rows}(\boldsymbol{C})}, \mathbf{c} = \frac{1}{\text{cols}(\boldsymbol{C})}$
Initialize dual variables: $\mathbf{u} = \mathbf{1}, \mathbf{v} = \mathbf{1}$
**for** $t = 1$ **to** $T$ **do**
    $\mathbf{u} = \mathbf{r} \oslash (\boldsymbol{K}\mathbf{v})$         ▷ $\oslash$ denotes element-wise division
    $\mathbf{v} = \mathbf{c} \oslash (\boldsymbol{K}^\top \mathbf{u})$
**end for**
**return** $\boldsymbol{P} = \text{diag}(\mathbf{u}) \cdot \boldsymbol{K} \cdot \text{diag}(\mathbf{v})$

---

the player explores previously unseen regions. Additionally, the inventory interface—fixed at the bottom of the screen—requires updates to token values without positional shifts. To accommodate these patterns, we apply the optimal transport output to the central region of the screen, where token reuse is most appropriate, while using the transformer's predictions for the screen edges and inventory regions, where new content is more likely. For MinAtar, which does not have special behavior at the edges, the optimal transport output is used directly for the entire screen.

**Choosing hyperparameters for optimal transport decoding** Two hyperparameters are introduced in optimal transport-based decoding: $c_d$, a coefficient for the distance cost, and $c_w$, a constant penalty for using a wildcard token. The best hyperparameters are chosen by grid search, but optimal transport-based decoding is robust to varying choices as shown in Tables 9 and 10.

---

**Algorithm 3** BINARIZATION of partial transport plan

---

**Input:** Partial transport plans $\boldsymbol{P}^{(prev)}$, $\boldsymbol{P}^{(gen)}$,
 large value $v$
**Output:** Binarized transport plans $\boldsymbol{\Pi}^{(prev)}$, $\boldsymbol{\Pi}^{(gen)}$

$\boldsymbol{P}^{\text{in}}$ = concatenate $\left(\boldsymbol{P}^{(prev)}, \boldsymbol{P}^{(gen)}\right)$
Initialize $\boldsymbol{P}^{(0)} = \boldsymbol{P}^{\text{in}}$, $t = 0$
**repeat**
    target = $\operatorname{argmax}(\boldsymbol{P}^{(t)}, \dim = 1)$
    $\boldsymbol{\Pi}^{\text{initial}} = \boldsymbol{0}_{n \times m}$
    **for** $i = 0$ to $n - 1$ **do**
        $\boldsymbol{\Pi}^{\text{initial}}[i, \text{target}[i]] = 1$
    **end for**
    $\boldsymbol{C} = \boldsymbol{P}^{(t)} \odot \boldsymbol{\Pi}^{\text{initial}} - v(1 - \boldsymbol{\Pi}^{\text{initial}})$
    source = $\operatorname{argmax}(\boldsymbol{C}, \dim = 0)$
    $\boldsymbol{\Pi}^{\text{out}} = \boldsymbol{0}_{n \times m}$
    **for** $j = 0$ to $m - 1$ **do**
        $\boldsymbol{\Pi}^{\text{out}}[\text{source}[j], j] = 1$
    **end for**
    $\boldsymbol{\Pi}^{\text{out}} = \boldsymbol{\Pi}^{\text{out}} \odot \boldsymbol{\Pi}^{\text{initial}}$
    $\boldsymbol{R} = (1 - \boldsymbol{\Pi}^{\text{out}}) \odot \boldsymbol{\Pi}^{\text{initial}}$
    $\boldsymbol{P}^{(t+1)} = \boldsymbol{P}^{(t)} - v\boldsymbol{R}$
    $t = t + 1$
**until** $\boldsymbol{\Pi}^{\text{out}} = \boldsymbol{\Pi}^{\text{initial}}$
$\boldsymbol{\Pi}^{(prev)} = \boldsymbol{\Pi}^{\text{out}}[1 : L, 1 : L]$
$\boldsymbol{\Pi}^{(gen)} = \boldsymbol{\Pi}^{\text{out}}[L + 1 : 2L, 1 : L]$
**return** $\boldsymbol{\Pi}^{(prev)}$, $\boldsymbol{\Pi}^{(gen)}$

---

Table 9: Average returns and scores with respect to $c_d$, a coefficient of cost for distance.

| $c_d$ | Return (%) | Score (%) |
|---|---|---|
| 0.0 | 72.08 | 32.89 |
| 0.3 | 71.10 | 31.41 |
| 0.6 | 72.46 | 35.60 |
| 0.8 | 71.49 | 33.54 |

# E  CRAFTAX DESCRIPTION AND HYPERPARAMETERS

Craftax is a complex environment inspired by Craftax-classic, featuring a larger screen, more enemies, more items, and multiple underground levels. An example observation is shown in Figure 11. Following Dedieu et al. (2025), we change some hyperparameters for Craftax, as shown in Table 11. In particular, to accommodate the larger screen and additional tokens in memory, the batch size and replay buffer size are reduced.

# F  MINATAR RETURN CURVES AND HYPERPARAMETERS

Figure 9 shows the return curves of TWISTED and baseline Dedieu et al. (2025) for each game in MinAtar. TWISTED outperforms the baseline in every game. Table 12 lists the hyperparameters for MinAtar with different values compared to Craftax-classic. All hyperparameter changes follow Dedieu et al. (2025), except the hyperparameters specific to optimal transport. Also following Dedieu et al. (2025), the policy encoder uses layer normalization and the Swish activation function, and actor and value networks share weights except in their final linear layers (Ba et al., 2016; Ramachandran et al., 2017).

Table 10: Average returns and scores with respect to $c_w$, a constant penalty for using a wildcard token.

| $c_w$ | Return (%) | Score (%) |
|---|---|---|
| 0.3 | 72.46 | 35.60 |
| 0.6 | 71.60 | 35.41 |

Table 11: Hyperparameter differences between Craftax-classic and Craftax.

| Area | Hyperparameter | Craftax-classic | Craftax |
|---|---|---|---|
| Environment | Observation shape | $63 \times 63 \times 3$ | $130 \times 110 \times 3$ |
| | Number of possible actions | 17 | 43 |
| | Number of possible achievements | 22 | 226 |
| | Number of environments (batch size) | 48 | 16 |
| Tokenizer | Single patch shape | $7 \times 7 \times 3$ | $10 \times 10 \times 3$ |
| Architecture | State tokens per frame $L$ | 81 | 143 |
| Training | Replay buffer size | $128,000$ | $48,000$ |

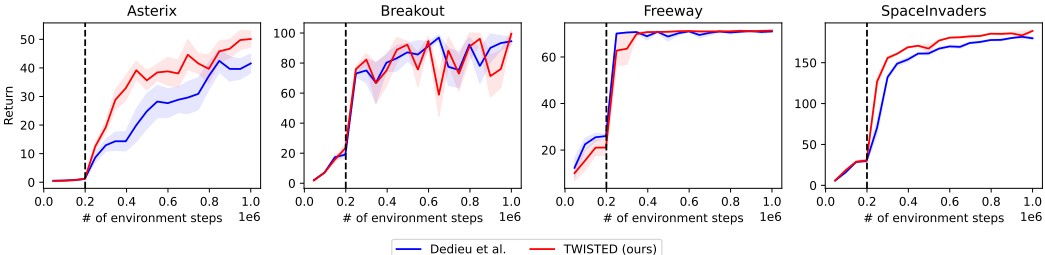

Figure 9: Return curves for MinAtar. Shading indicates standard error among multiple seeds. The vertical dashed lines indicate the start of training in imagination after $T_{\text{warmup}}$ interactions.

## G COMPARISON TO $\Delta$-IRIS

To further evaluate the impact of optimal transport-based decoding, we compare against $\Delta$-IRIS, a previous approach that also seeks to exploit visual redundancy between frames (Micheli et al., 2024). We apply the $\Delta$-IRIS approach to the TWISTED architecture, by replacing the nearest-neighbor tokenizer with $\Delta$-IRIS's tokenizer that encodes the delta difference between frames into tokens. We evaluate this model without OT, to compare the impact of $\Delta$-IRIS vs. OT. Table 13 shows the return and score of $\Delta$-IRIS applied to TWISTED compared to TWISTED on Craftax-classic with 1M interactions. $\Delta$-IRIS without OT performs worse than TWISTED, showing that OT is better at exploiting visual redundancies between frames than the $\Delta$-IRIS method on our architecture.

## H COMPUTE TIME

Table 14 reports the running time of TWISTED, TWISTED with spatio-temporal positional encoding disabled (OT only), and its baseline Dedieu et al. (2025) using the same hyperparameters. Optimal transport-based decoding increases the overall end-to-end training time by only 2.8%. Spatio-temporal positional encoding introduces neglible overhead, increasing world model training time by only 1%.

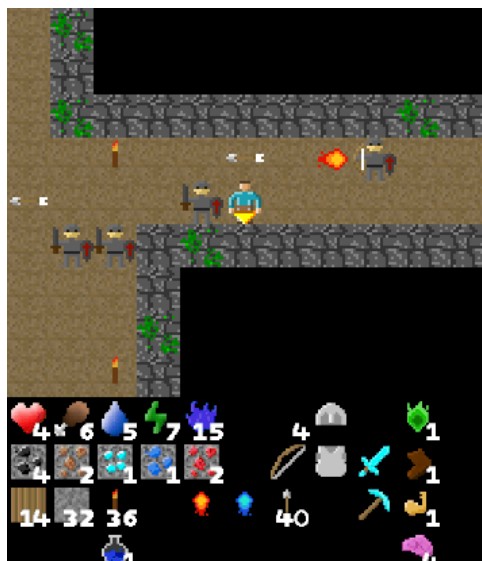

Figure 8: An example observation of Craftax.

Table 12: Hyperparameter differences between Craftax-classic and MinAtar. $K$ is the number of object types for each game (4 for Asterix, 4 for Breakout, 7 for Freeway, and 6 for SpaceInvaders). The number of actions $A$ is 5 for Asterix, 3 for Breakout, 3 for Freeway, and 4 for SpaceInvaders.

| Area | Hyperparameter | Craftax-classic | MinAtar |
|---|---|---|---|
| Environment | Observation shape | $63 \times 63 \times 3$ | $10 \times 10 \times K$ |
| | Number of possible actions | 17 | $A$ |
| | Warmup interactions $T_{\text{warmup}}$ | 50,000 | 200,000 |
| Tokenizer | Single patch shape | $7 \times 7 \times 3$ | $2 \times 2 \times K$ |
| Architecture | State tokens per frame $L$ | 81 | 25 |
| Optimal transport | Distance cost coefficient $c_d$ | 0.6 | 0.2 |
| | Wildcard cost $c_w$ | 0.3 | 0.05 |
| Training | Number of world model updates $U_{\text{WM}}$ | 500 | 2000 |
| | Number of policy updates in imagination $U_{\text{imag}}$ | 300 | 2000 |
| | Coefficient for reward prediction loss | 1 | 10 |
| | Coefficient for done prediction loss | 1 | 10 |
| PPO | Discount factor $\gamma$ | 0.925 | 0.95 |
| | TD weight $\lambda$ | 0.625 | 0.75 |
| | Entropy loss coefficient $\lambda_{\text{ent}}$ in imagination | 0.01 | 0.05 |
| | PPO target discount factor $\alpha$ | 0.95 | 0.925 |

## I    ADDITIONAL ATARI 100K RESULTS

Table 15 shows the average return for each of 26 games in Atari 100K.

Table 13: Comparison of applying $\Delta$-IRIS to TWISTED without OT, on Craftax-classic after 1M environment interactions.

| Method | Return (%) | Score (%) |
|---|---|---|
| TWISTED without OT + $\Delta$-IRIS | $28.77 \pm 0.67$ | $5.84 \pm 0.31$ |
| TWISTED (ours) | $\mathbf{72.46} \pm 0.45$ | $\mathbf{35.60} \pm 0.92$ |

Table 14: Running times on a single Nvidia RTX 3090 GPU. WM training measures one epoch of world model training. Imagination measures one epoch of policy training in imagination. Total time represents end-to-end training time for 1M environment steps.

| Method | WM training (m) | Imagination (m) | Total time (hrs) |
|--------|-----------------|-----------------|------------------|
| Baseline | 8.08 | 4.48 | 46.3 |
| OT only | 8.09 | 4.76 | 47.6 |
| TWISTED (ours) | 8.19 | 4.78 | 48.2 |

Table 15: Mean returns on the 26 games of the Atari 100k benchmark followed by averaged human-normalized performance metrics. Each game score is computed as the average of 5 runs with different seeds. Bold face mark the best score.

| Game | Random | Human | DreamerV3 | STORM | DIAMOND | SIMULUS (reproduced) | TWISTED (ours) |
|------|--------|-------|-----------|-------|---------|----------------------|----------------|
| Alien | 227.8 | 7127.7 | 959.4 | **983.6** | 744.1 | 734.3 | 727.4 |
| Amidar | 5.8 | 1719.5 | 139.1 | 204.8 | **225.8** | 113.5 | 144.7 |
| Assault | 222.4 | 742.0 | 705.6 | 801.0 | 1526.4 | **1652.2** | 1455.2 |
| Asterix | 210.0 | 8503.3 | 932.5 | 1028.0 | **3698.5** | 1399.5 | 1610.2 |
| BankHeist | 14.2 | 753.1 | **648.7** | 641.2 | 19.7 | 504.6 | 370.6 |
| BattleZone | 2360.0 | 37187.5 | 12250.0 | 13540.0 | 4702.0 | 6792.3 | **13590.4** |
| Boxing | 0.1 | 12.1 | 78.0 | 79.7 | 86.9 | 90.9 | **91.9** |
| Breakout | 1.7 | 30.5 | 31.1 | 15.9 | 132.5 | **167.4** | 75.4 |
| ChopperCommand | 811.0 | 7387.8 | 410.0 | 1888.0 | 1369.8 | 3930.6 | **4720.8** |
| CrazyClimber | 10780.5 | 35829.4 | 97190.0 | 66776.0 | **99167.8** | 86021.3 | 95114.2 |
| DemonAttack | 152.1 | 1971.0 | 303.3 | 164.6 | 288.1 | **4863.4** | 4814.6 |
| Freeway | 0.0 | 29.6 | 0.0 | 0.0 | **33.3** | 32.4 | 32.0 |
| Frostbite | 65.2 | 4334.7 | 909.4 | **1316.0** | 274.1 | 260.5 | 266.5 |
| Gopher | 257.6 | 2412.5 | 3730.0 | **8239.6** | 5897.9 | 3815.2 | 4967.8 |
| Hero | 1027.0 | 30826.4 | **11160.5** | 11044.3 | 5621.8 | 7531.1 | 6603.0 |
| Jamesbond | 29.0 | 302.8 | 444.6 | 509.0 | 427.4 | 664.9 | **1126.9** |
| Kangaroo | 52.0 | 3035.0 | 4098.0 | 4208.0 | 5382.2 | **9918.7** | 9725.4 |
| Krull | 1598.0 | 2665.5 | 7781.5 | 8412.6 | **8610.1** | 7178.5 | 6446.7 |
| KungFuMaster | 258.5 | 22736.3 | 21420.0 | **26182.0** | 18713.6 | 18581.2 | 19948.8 |
| MsPacman | 307.3 | 6951.6 | 1326.9 | **2673.5** | 1958.2 | 1094.0 | 1133.7 |
| Pong | -20.7 | 14.6 | 18.4 | 11.3 | **20.4** | 17.5 | 18.5 |
| PrivateEye | 24.9 | 69571.3 | 881.6 | **7781.0** | 114.3 | 100.0 | 726.3 |
| Qbert | 163.9 | 13455.0 | 3405.1 | **4522.5** | 4499.3 | 3176.5 | 3194.6 |
| RoadRunner | 11.5 | 7845.0 | 15565.0 | 17564.0 | 20673.2 | 22177.7 | **23535.4** |
| Seaquest | 68.4 | 42054.7 | 618.0 | 525.2 | 551.2 | **1727.2** | 1273.8 |
| UpNDown | 533.4 | 11693.2 | 7567.1 | 7985.0 | 3856.3 | 4356.3 | **12901.1** |
| #Superhuman (↑) | 0 | N/A | 9 | 9 | 11 | 12 | **13** |
| Mean (↑) | 0.000 | 1.000 | 1.124 | 1.222 | 1.459 | **1.636** | 1.616 |
| Median (↑) | 0.000 | 1.000 | 0.485 | 0.425 | 0.373 | 0.739 | **0.978** |
| IQM (↑) | 0.000 | 1.000 | 0.487 | 0.561 | 0.641 | 0.969 | **1.092** |
| Optimality Gap (↓) | 1.000 | 0.000 | 0.510 | 0.472 | 0.480 | 0.410 | **0.376** |

