# OpenReview forum: "TWISTED: Enhancing Transformer World Models with Spatio-Temporal Encoding and Graph-Based Optimal Decoding"
_ICLR.cc/2026/Conference — Submitted to ICLR 2026_

### Official Review · Reviewer_p88M · 2025-10-22

**Soundness:** 3
**Presentation:** 4
**Contribution:** 3
**Rating:** 6
**Confidence:** 4

**Summary:**

This paper presents TWISTED, a transformer-based world model that integrates richer spatial and temporal structure via (1) a **3D spatio-temporal positional encoding** scheme and (2) a **graph-based optimal transport (OT) decoding mechanism**. The first component generalizes standard 1D positional encodings to better handle video or spatially grounded RL tasks, while the second aligns generated tokens across time using OT, reducing temporal inconsistency and object drift. The model achieves improved performance on several visual RL benchmarks, including Craftax-classic and MinAtar, showing better stability and long-horizon prediction accuracy.

**Strengths:**

* The motivation is well-grounded: visual world models naturally involve spatial-temporal relationships, and standard RoPE encodings cannot represent them adequately.

* The proposed OT decoding is both novel and intuitive, promoting temporal coherence by reusing tokens from prior frames via transport mapping.

* Experiments show consistent improvement across datasets, and qualitative samples convincingly illustrate more stable rollouts.

* The writing style is clear, with excellent diagrams explaining both components.

* The method’s modularity allows easy integration with other transformer backbones.

**Weaknesses:**

* The ablation section is somewhat limited; it does not isolate the effects of spatio-temporal encoding and OT decoding separately.

* Computational costs (both training and inference) for the OT solver are not analyzed, which may be substantial in practice.

* Comparisons are missing with recent state-space world models (e.g., Mamba, S4WM) that offer long-context handling with lower overhead.

* There is little theoretical analysis of OT stability under noisy or high-variance feature representations.

**Questions:**

1. Could you provide ablations that test encoding-only and decoding-only contributions?

2. Which OT solver and regularization settings (e.g., ε, Sinkhorn iterations) were used, and how sensitive are results to them?

3. How much additional computational cost does OT decoding introduce?

4. Can this approach scale to higher-resolution visual inputs?

---

> ### Author Response · Authors · 2025-11-21
>
> Thank you very much for your review. We appreciate that you found TWISTED to be well-motivated and the quantitative and qualitative results to be convincing.
>
>
> ---
> > Weakness 1: The ablation section is somewhat limited; it does not isolate the effects of spatio-temporal encoding and OT decoding separately.
>
> The effects of spatio-temporal encoding and OT decoding are isolated in Table 2. The “STPE only” row shows the result of using only STPE without any involvement of OT, and vice versa for the “Optimal transport only” row.
>
> \
> *Table 2: Ablations on Craftax-classic with 1M interactions.*
>
> | Method                 | Return (\%)          | Score (\%)           |
> |------------------------|----------------------|----------------------|
> | Dedieu et al. (2025)   | 68.55 $\pm$ 0.72     | 27.24 $\pm$ 0.86     |
> | Relative PE only       | 69.26 $\pm$ 0.50     | 29.37 $\pm$ 0.80     |
> | STPE only              | 71.85 $\pm$ 0.63     | 33.94 $\pm$ 1.10     |
> | Optimal transport only | 69.77 $\pm$ 0.65     | 31.08 $\pm$ 0.88     |
> | TWISTED (ours)          | **72.46** $\pm$ 0.45 | **35.60** $\pm$ 0.92 |
>
> ---
> > Weakness 2: Computational costs (both training and inference) for the OT solver are not analyzed
>
> We have updated Appendix H to include the computational cost of only OT, without the effects of STPE. The computational cost of OT is not substantial; overall computation time only increases by 2.8% (Table 14). The OT solver is not used during world model training; it is only used during world model inference for policy training in imagination (see the “Imagination” column).
>
> \
> *Table 14: Running times on a single Nvidia RTX 3090 GPU.
> WM training measures one epoch of world model training. Imagination measures one epoch of policy training in imagination. Total time represents end-to-end training time for 1M environment steps.*
>
> | Method         | WM training (m) | Imagination (m) | Total time (hrs) |
> |----------------|-----------------|-----------------|------------------|
> | Baseline       | 8.08            | 4.48            | 46.3             |
> | OT only        | 8.09            | 4.76            | 47.6             |
> | TWISTED (ours) | 8.19            | 4.78            | 48.2             |
>
> ---
> > Weakness 3: Comparisons are missing with recent state-space world models (e.g., Mamba, S4WM)
>
> We compare with the state-of-the-art state-space world model for the Crafter benchmark, which is DreamerV3. We do not directly compare with Mamba and S4WM because they do not report results for our benchmarks.
>
> ---
> > Weakness 4: There is little theoretical analysis of OT stability under noisy or high-variance feature representations.
>
> In practice, OT remains stable for realistic feature representations encountered in RL environments, as demonstrated empirically across multiple environments and in the qualitative analysis. Thus, theoretical analysis of OT stability under noisy or high-variance feature representations would be tangential to this paper.

---

> ### Author Response · Authors · 2025-11-21
>
> > Question 1: Could you provide ablations that test encoding-only and decoding-only contributions?
>
> STPE is encoding-only, and its independent contribution is shown in the “STPE only” row of Table 2. OT is decoding-only, and its contribution is shown in the “Optimal transport only” row of Table 2.
>
> \
> *Table 2: Ablations on Craftax-classic with 1M interactions.*
>
> | Method                 | Return (\%)          | Score (\%)           |
> |------------------------|----------------------|----------------------|
> | Dedieu et al. (2025)   | 68.55 $\pm$ 0.72     | 27.24 $\pm$ 0.86     |
> | Relative PE only       | 69.26 $\pm$ 0.50     | 29.37 $\pm$ 0.80     |
> | STPE only              | 71.85 $\pm$ 0.63     | 33.94 $\pm$ 1.10     |
> | Optimal transport only | 69.77 $\pm$ 0.65     | 31.08 $\pm$ 0.88     |
> | TWISTED (ours)          | **72.46** $\pm$ 0.45 | **35.60** $\pm$ 0.92 |
>
> ---
> > Question 2: Which OT solver and regularization settings (e.g., ε, Sinkhorn iterations) were used, and how sensitive are results to them?
>
> We used the OTT-JAX library implementation of the Sinkhorn solver[1]. As reported in Table 7, we chose a relative $\epsilon$ of 0.00001, relative to the standard deviation of the cost matrix, and 10 Sinkhorn iterations, after we found that the results are not sensitive to varying Sinkhorn iterations.
>
> \
> *Excerpt of Table 7: World model hyperparameters. Sweep range indicates the values tried per hyperparameter, with the final Value being chosen based on highest return.*
>
> | Area              | Hyperparameter                               | Value   | Sweep range            |
> |-------------------|----------------------------------------------|---------|------------------------|
> | Optimal transport | Distance cost coefficient $c_d$              | 0.6     | \{0.0, 0.3, 0.6, 0.8\} |
> |                   | Wildcard cost $c_w$                          | 0.3     | \{0.3, 0.6\}           |
> |                   | Sinkhorn regularization parameter $\epsilon$ | 0.00001 | \{0.00001, 0.0001\}    |
> |                   | Sinkhorn iterations                          | 10      | \{10, 100, 500\}       |
>
> ---
> > Question 3: How much additional computational cost does OT decoding introduce?
>
>
> OT decoding only adds 2.8% of computational cost, as shown in the updated Table 14.
>
> \
> *Table 14: Running times on a single Nvidia RTX 3090 GPU.
> WM training measures one epoch of world model training. Imagination measures one epoch of policy training in imagination. Total time represents end-to-end training time for 1M environment steps.*
>
> | Method         | WM training (m) | Imagination (m) | Total time (hrs) |
> |----------------|-----------------|-----------------|------------------|
> | Baseline       | 8.08            | 4.48            | 46.3             |
> | OT only        | 8.09            | 4.76            | 47.6             |
> | TWISTED (ours) | 8.19            | 4.78            | 48.2             |
>
> ---
> > Question 4: Can this approach scale to higher-resolution visual inputs?
>
>
> Yes, this approach can scale to higher-resolution visual inputs because the computational costs of STPE and OT are not significant.
>
> ---
> [1] Cuturi et al. “Optimal Transport Tools (OTT): A JAX Toolbox for all things Wasserstein.” arXiv preprint arXiv:2201.12324, 2022.

---

### Official Review · Reviewer_Niit · 2025-10-23

**Soundness:** 2
**Presentation:** 2
**Contribution:** 1
**Rating:** 2
**Confidence:** 4

**Summary:**

The paper proposes two components for improving the performance of transformer world models in visual environments: one addresses temporal and spatial adjacency / positional information encoding and the other addresses the significant visual redundancy in certain visual applications.
The method is evaluated on the Craftax and MinAtar benchmarks, presenting improved performance compared to existing results.

**Strengths:**

- The paper is overall well-written and easy to follow.
- The ablations setup and results are informative and well executed. The experimental design and results allow to draw meaningful conclusions on the impact of each of the proposed components.
- The proposed optimal transport decoding mechanism is original.
- The approach, TWISTED, outperforms previous baselines on the Craftax, Craftax-Classic and MinAtar benchmarks.

**Weaknesses:**

`W1`: The spatio-temporal position encoding (STPE) approach is very similar to methods proposed in several prior works [1][2][3][4], under the name 3D RoPE, even in the context of world models, including the combination of 3D RoPE and learned absolute positional encoding [1].

Comparing (extensively) the proposed STPE to VideoRoPE and the above 3D RoPE methods would be necessary to support the necessity of STPE.

Notably, according to the ablations study (Table 2), the most significant contributor to the final performance is the spatio-temporal position encoding mechanism (STPE).



`W2` Similarly, in the related works section, under "Positional embeddings in video modeling", 3D RoPE methods were not sufficiently covered.


`W3`: The second claimed contribution is the optimal transport decoding, which addresses the problem of poor utilization of the significant visual redundancy in some video applications.
1. Given the empirical evidence in the paper (Table 2), its contribution to the final performance is questionable, and marginal at best.
2. This component adds a non-trivial complexity to the overall method.
3. In contrast to prior approaches such as Delta-IRIS, which ultimately yields shorter token sequences and thus benefits improved efficiency (lower computational cost), the proposed approach *adds* a computational cost on top of that of the baseline.
4. Since the proposed algorithm, TWISTED, is the only baseline that uses this positional encoding mechanism, it is unclear how prior works such as delta IRIS would perform under the implementation of TWISTED, in other words, whether the observed improvement (if not a statistical fluctuation) stems primarily from 3D-RoPE + implementation and architectural differences.

To support this approach empirically, it is required to consider an experimental setup for studying solutions to the poor utilization of visual redundancy problem in isolation.

To evaluate this proposed approach more reliably, I would suggest to consider the following setup:
Starting from a baseline version of TWISTED where optimal transport decoding is disabled, add the Delta-IRIS mechanism. Similarly, consider other appropriate baselines as well. Then, evaluate all baselines, and compare the performance to those of TWISTED.

This would allow to eliminate any external factor such as implementation details and the advantage of 3D RoPE, and would show the impact of the proposed component in isolation.



`W4`:
> In this paper, we present TWISTED, a transformer world model tailored for visual RL environments. (line 480)

The paper claims that the method addresses visual RL environments (a very general claim). However, in practice, the focus is on visual environments with significant visual redundancy such as Atari and Craftax, where a significant overlap exists between many frames. In more complex visual environments, such as Minecraft, autonomous driving, or real-world robotics, it is unclear whether such redundancy is significant enough to make a difference in performance, and whether the method would maintain performance in cases with insignificant frame overlap (in the context of the optimal transport decoding). Since the paper does not provide supporting evidence for such environments, I believe the scope of the claims should be narrowed.


`W5`: The paper lacks an explicit formal description of the STPE method (in mathematical notation).


`W6`: It would be valuable to extend the ablations to MinAtar as well.

--------

[1] Agarwal, N., Ali, A., Bala, M., Balaji, Y., Barker, E., Cai, T., ... & Zolkowski, A. (2025). Cosmos world foundation model platform for physical ai. arXiv preprint arXiv:2501.03575.

[2] Song, K., Chen, B., Simchowitz, M., Du, Y., Tedrake, R., & Sitzmann, V. (2025). History-guided video diffusion. arXiv preprint arXiv:2502.06764.

[3] Yang, Z., Teng, J., Zheng, W., Ding, M., Huang, S., Xu, J., ... & Tang, J. CogVideoX: Text-to-Video Diffusion Models with An Expert Transformer. In The Thirteenth International Conference on Learning Representations.

[4] Kong, W., Tian, Q., Zhang, Z., Min, R., Dai, Z., Zhou, J., ... & Zhong, C. (2024). Hunyuanvideo: A systematic framework for large video generative models. arXiv preprint arXiv:2412.03603.

**Questions:**

`Q1`:
> Return is averaged over episodes of the final 50,000 environment interactions to smooth out variance (line 325)

A better practice would be to evaluate the final performance (of the fixed trained model) by collecting $K$ test episodes with the trained agent (fixed weights) and aggregate the resulting returns, with e.g., $K=100$ (per random seed).


`Q2`: In the related works section, you mentioned VideoRoPE, but did not specify how this approach differs from the proposed STPE, or justify why STPE is needed despite the existence of VideoRoPE. Did you try using VideoRoPE? What is missing in VideoRoPE that STPE offers? Are there empirical evidence to support this? Again, I would suggest including a comparison.


`Q3`: If I understand correctly, the tokenizer (Section 3.1) is not part of the novelty of the paper. Hence, it is more appropriate to include this information in the preliminaries section.

---

> ### Author Response · Authors · 2025-11-21
>
> Thank you very much for your detailed and thoughtful feedback. We’re glad that you find optimal transport-based decoding to be original and the state-of-the-art results to be meaningful and well-executed.
>
> ---
> > W1: The spatio-temporal position encoding (STPE) approach is very similar to methods proposed in several prior works
>
> 3D RoPE methods for video world models such as VideoRoPE[1] and Cosmos[2] are not adapted to the RL context because they do not handle spatio-temporal positional encoding for actions. In contrast, our proposed STPE approach introduces a 3D RoPE method to RL for the first time by giving a spatio-temporal positional encoding for actions that naturally associate them closely with the center of their preceding and succeeding state frames.
>
> ---
> > W2: Similarly, in the related works section, under "Positional embeddings in video modeling", 3D RoPE methods were not sufficiently covered.
>
> We have added [2][3][4][5] to the Related Works section.
>
> ---
> > W3: Given the empirical evidence in the paper (Table 2), [OT’s] contribution to the final performance is questionable, and marginal at best.
> > ... This component adds a non-trivial complexity to the overall method.
> > ... I would suggest to consider the following setup: Starting from a baseline version of TWISTED where optimal transport decoding is disabled, add the Delta-IRIS mechanism.
>
> The ablations and standard error intervals in Table 2 show that “Optimal transport only” makes a significant improvement by itself, and firmly contributes to the final performance above “STPE only”. OT does not add significant complexity to the method and only increases the end-to-end computational cost by 2.8% (Table 14).
>
> We have added an experiment comparing to $\Delta$-IRIS based on your suggestion (Table 13 in Appendix G). We added the $\Delta$-IRIS mechanism to the TWISTED architecture with OT disabled. This model performs worse than TWISTED, showing that OT exploits visual redundancy better than $\Delta$-IRIS and that TWISTED’s performance is not solely dependent on STPE and architectural changes.
>
> \
> *Table 2: Ablations on Craftax-classic with 1M interactions.*
>
> | Method                 | Return (\%)          | Score (\%)           |
> |------------------------|----------------------|----------------------|
> | Dedieu et al. (2025)   | 68.55 $\pm$ 0.72     | 27.24 $\pm$ 0.86     |
> | Relative PE only       | 69.26 $\pm$ 0.50     | 29.37 $\pm$ 0.80     |
> | STPE only              | 71.85 $\pm$ 0.63     | 33.94 $\pm$ 1.10     |
> | Optimal transport only | 69.77 $\pm$ 0.65     | 31.08 $\pm$ 0.88     |
> | TWISTED (ours)          | **72.46** $\pm$ 0.45 | **35.60** $\pm$ 0.92 |
>
> \
> *Table 14: Running times on a single Nvidia RTX 3090 GPU.
> WM training measures one epoch of world model training. Imagination measures one epoch of policy training in imagination. Total time represents end-to-end training time for 1M environment steps.*
>
> | Method         | WM training (m) | Imagination (m) | Total time (hrs) |
> |----------------|-----------------|-----------------|------------------|
> | Baseline       | 8.08            | 4.48            | 46.3             |
> | OT only        | 8.09            | 4.76            | 47.6             |
> | TWISTED (ours) | 8.19            | 4.78            | 48.2             |
>
> \
> *Table 13: Comparison of applying $\Delta$-IRIS to TWISTED without OT, on Craftax-classic after 1M environment interactions.*
> | Method                           | Return (\%)          | Score (\%) |
> |----------------------------------|----------------------|----------------------|
> | TWISTED without OT + $\Delta$-IRIS | 28.77 $\pm$ 0.67 | 5.84 $\pm$ 0.31   |
> | TWISTED (ours)                   | **72.46** $\pm$ 0.45 | **35.60** $\pm$ 0.92 |
>
>
> ---
> > W4: The paper claims that the method addresses visual RL environments (a very general claim).
>
> We have reworded the paper to specify that it focuses on “visual grid-based RL environments”, in line with prior work on transformer world models in grid-based environments [6].
>
> ---
> > W5: The paper lacks an explicit formal description of the STPE method (in mathematical notation).
>
> We have added a mathematical description of STPE in Appendix C.
>
> ---
> > W6: It would be valuable to extend the ablations to MinAtar as well.
>
> We believe our ablations on Craftax-classic are sufficient and ablations on MinAtar would not provide additional information. However, we plan to extend the ablations to MinAtar.

---

> ### Author Response · Authors · 2025-11-21
>
> > Q1: A better practice would be to evaluate the final performance (of the fixed trained model) by collecting $K$ test episodes.
>
> We report return as the average over final environment interactions as established in the original Crafter paper[7] and other Crafter baselines[8]. We follow the standard evaluation protocol on the Crafter benchmark so that our results can be compared to previously reported results.
>
> ---
> > Q2: Did you try using VideoRoPE? What is missing in VideoRoPE that STPE offers?
>
> VideoRoPE does not support spatio-temporal positional encoding for actions, so it cannot be directly applied in a reinforcement learning context. In contrast, our proposed STPE provides a spatio-temporal positional encoding for actions that naturally associate them closely with the preceding and succeeding states.
>
> We first tried adapting VideoRoPE to our setting by adding encoding for actions. However, this adapted form of VideoRoPE is still not sufficient (see “Relative PE only” in Table 2). Instead, our proposed STPE also includes absolute positional embeddings, which significantly improve performance in our RL setting (“STPE only” vs. “Relative PE only” in Table 2).
>
> \
> *Table 2: Ablations on Craftax-classic with 1M interactions.*
>
> | Method                 | Return (\%)          | Score (\%)           |
> |------------------------|----------------------|----------------------|
> | Dedieu et al. (2025)   | 68.55 $\pm$ 0.72     | 27.24 $\pm$ 0.86     |
> | Relative PE only       | 69.26 $\pm$ 0.50     | 29.37 $\pm$ 0.80     |
> | STPE only              | 71.85 $\pm$ 0.63     | 33.94 $\pm$ 1.10     |
> | Optimal transport only | 69.77 $\pm$ 0.65     | 31.08 $\pm$ 0.88     |
> | TWISTED (ours)          | **72.46** $\pm$ 0.45 | **35.60** $\pm$ 0.92 |
>
> ---
> > Q3: If I understand correctly, the tokenizer (Section 3.1) is not part of the novelty of the paper. Hence, it is more appropriate to include this information in the preliminaries section.
>
> We have moved the tokenizer to the Preliminaries section (Section 2.4).
>
> ---
> [1] Wei et al. “VideoRoPE: What Makes for Good Video Rotary Position Embedding?”. arXiv preprint arXiv:2502.05173, 2025.
>
> [2] Agarwal et al. “Cosmos World Foundation Model Platform for Physical AI”. arXiv preprint arXiv:2501.03575, 2025.
>
> [3] Song et al. “History-guided Video Diffusion”. ICML, 2025.
>
> [4] Yang et al. “CogVideoX: Text-to-Video Diffusion Models with An Expert Transformer”. ICLR, 2025.
>
> [5] Kong et al. “Hunyuanvideo: A systematic framework for large video generative models”. arXiv preprint arXiv:2412.03603, 2024.
>
> [6] Dedieu et al. “Improving Transformer World Models for Data-Efficient RL.” ICML, 2025.
>
> [7] Hafner. “Benchmarking the Spectrum of Agent Capabilities.” ICLR, 2022.
>
> [8] Moon et al. “Discovering Hierarchical Achievements in Reinforcement Learning via Contrastive Learning.” NeurIPS, 2023.

---

> > ### Comment · Reviewer_Niit · 2025-11-26
> >
> > I appreciate the author's detailed and thoughtful response, and their willingness to improve the paper within the discussion timeframe.
> >
> > #### STPE Novelty
> >
> > While the authors argue in their rebuttal that the novelty of their STPE method lies in the explicit handling of action tokens, I find this argument inconsistent with the overall story and presentation in the paper itself. Notably, the abstract and introduction tell a different story.
> >
> > If the novelty of the method lies in the handling of actions, I would expect all 3D RoPE and non-action embedding details to be presented as preliminaries. Furthermore, I would expect the experiments and ablations to focus on the action embeddings specifically. For example, to consider even a simple or trivial baseline for the handling actions with 3D RoPE.
> > In this regard, such a baseline could be to apply only the temporal embedding for actions tokens, or to use a fixed spatial value.
> >
> > The related work section could also have a specific focus on the action aspect.
> >
> >
> >
> > ------
> >
> >
> >
> > > Line 470: However, these video-based 3D RoPE methods cannot be directly applied in an RL context because they do not support spatio-temporal embeddings for actions
> >
> > To be fair, these methods can in fact be applied in an RL context, if actions are represented in other ways. For example, observation embeddings may be mapped in a way that also encodes related actions.
> >
> >
> >
> >
> > #### Optimal Transport
> >
> > Thank you for the additional evidence provided to support the optimal transport component. However, I find the reported performance (28.77) concerning, as it is substantially lower than even the original baseline results (68.55; Dedieu et al., 2025). In contrast, the Delta IRIS paper reported maintained or improved performance. This discrepancy raises the possibility of an error, which may understandably occur under the time constraints and pressure of the rebuttal period.
> >
> >
> >
> >
> >
> > Overall, the proposed optimal transport component is interesting, but I continue to have concerns regarding its empirical impact. Furthermore, the inconsistency between the main storyline and the action embedding method argument weakens the overall coherence. In its present form, I feel the paper does not yet meet the standard for acceptance.

---

> > > ### Author Response · Authors · 2025-11-27
> > >
> > > Thank you for your follow-up response.
> > >
> > > > However, I find the reported performance (28.77) concerning, as it is substantially lower than even the original baseline results (68.55; Dedieu et al., 2025).
> > >
> > > Note that the reported return in the original Delta IRIS paper is 35.00, which is substantially lower than the Dedieu baseline result (68.55). While Delta IRIS improves on IRIS, there is no reason to believe that Delta IRIS will increase the Dedieu baseline score when applied to it. A disadvantage of Delta IRIS is that it requires replacing a model’s tokenizer with Delta IRIS’s VQ-VAE tokenizer in order to apply its method. The Dedieu baseline is optimized for its own nearest neighbor tokenizer, so it is expected that Delta IRIS will not perform as well when applied to Dedieu.
> > >
> > > Our results show that Delta IRIS does not generalize to other architectures such as Dedieu, which rely on a different tokenizer than Delta IRIS's tokenizer. In contrast, our optimal transport-based decoding method can be readily applied to recent baselines, by targeting the underexplored area of decoding strategy instead of replacing the tokenizer, adding additional improvement on top of world models using nearest-neighbor tokenizers.
> > >
> > > > This discrepancy raises the possibility of an error, which may understandably occur under the time constraints and pressure of the rebuttal period.
> > >
> > > We have carefully tested our implementation of Delta IRIS for correctness. We have updated the supplementary material with our Delta IRIS code, if reviewers would like to review it.

---

> ### Author Response · Authors · 2025-12-03
>
> > W4: The paper claims that the method addresses visual RL environments (a very general claim).
>
> Thank you for suggesting more complex visual environments to demonstrate the generality of TWISTED and strengthen our paper.
>
> We have added experiments on Atari 100K, which has significantly less visual redundancy between frames for games such as BattleZone, ChopperCommand, and Hero. Our state-of-the-art performance on Atari 100K demonstrates the generality of TWISTED across visual RL environments. However, our paper’s focus remains on exploiting visual redundancy between frames, which is a widespread phenomenon among popular RL benchmarks such as Atari and Craftax. Thus, we have reworded the paper to specify that it focuses on “2D visual RL environments”, rather than aiming for environments like autonomous driving or real-world robotics.
>
> \
> *Table 6: Aggregate metrics on Atari 100K after 100K environment interactions. Return for each game is evaluated on 100 evaluation episodes at the end of training.*
> |                      | IQM (↑) | Optimality Gap (↓) | Mean (↑) | Median(↑) |
> |----------------------|---------|--------------------|----------|-----------|
> | Simulus (reproduced) |   0.969 |              0.410 |    **1.636** |     0.739 |
> | TWISTED (ours)       |   **1.092** |              **0.376** |    1.616 |     **0.978** |

---

### Official Review · Reviewer_tLB5 · 2025-10-28

**Soundness:** 3
**Presentation:** 4
**Contribution:** 3
**Rating:** 6
**Confidence:** 4

**Summary:**

This paper proposes TWISTED, a transformer-based world model that incorporates two main components: a 3D spatio-temporal positional encoding and an optimal transport-based decoding mechanism. The method is evaluated on several challenging benchmarks (Craftax-classic, Craftax, MinAtar), achieving state-of-the-art performance. The core idea of using optimal transport to enforce object persistence across frames is interesting and well-motivated for visual environments.

**Strengths:**

+ The optimal transport (OT) based decoding is novel and well-motivated.
+ The 3D spatio-temporal positional encoding (STPE) in world modeling is novel and effective.
+ The empirical evaluation is comprehensive and convincing, demonstrating SOTA performance.

**Weaknesses:**

- The paper presents two independent ideas (STPE and OT). Ablation studies show both help, yet it is not clear whether the performance gain is additive or synergistic. It is missing a baseline combining STPEwith a simpler and less expensive decoding heuristic.
- The authors should provide a detailed breakdown of the OT's cost and discuss the scalability of their Sinkhorn solution to environments with more tokens (e.g., higher-resolution images).
- The OT formulation uses fixed cost coefficients (c_d, c_w) and it is not disscused how they are selected and how sensitive are the results.

**Questions:**

See above.

---

> ### Author Response · Authors · 2025-11-21
>
> Thank you very much for your review and your feedback. We’re glad that you are convinced by our experiments demonstrating SOTA performance and the novelty and effectiveness of our method.
>
> ---
> > Weakness 1: Ablation studies show both [STPE and OT] help, yet it is not clear whether the performance gain is additive or synergistic. It is missing a baseline combining STPE with a simpler and less expensive decoding heuristic.
>
> The ablations in Table 2 show that the performance gain of STPE and OT are additive, since “STPE only” and “Optimal transport only” both make significant improvement over the baseline (rather than only their combination making a synergistic improvement). The “STPE only” ablation demonstrates STPE with a simple decoding method, namely decoding all tokens from the transformer output, as is standard in baselines. Note that OT is not an expensive decoding method, as it only adds 2.8% in end-to-end running time (Table 14).
>
> \
> *Table 2: Ablations on Craftax-classic with 1M interactions.*
>
> | Method                 | Return (\%)          | Score (\%)           |
> |------------------------|----------------------|----------------------|
> | Dedieu et al. (2025)   | 68.55 $\pm$ 0.72     | 27.24 $\pm$ 0.86     |
> | Relative PE only       | 69.26 $\pm$ 0.50     | 29.37 $\pm$ 0.80     |
> | STPE only              | 71.85 $\pm$ 0.63     | 33.94 $\pm$ 1.10     |
> | Optimal transport only | 69.77 $\pm$ 0.65     | 31.08 $\pm$ 0.88     |
> | TWISTED (ours)          | **72.46** $\pm$ 0.45 | **35.60** $\pm$ 0.92 |
>
> \
> *Table 14: Running times on a single Nvidia RTX 3090 GPU.
> WM training measures one epoch of world model training. Imagination measures one epoch of policy training in imagination. Total time represents end-to-end training time for 1M environment steps.*
>
> | Method         | WM training (m) | Imagination (m) | Total time (hrs) |
> |----------------|-----------------|-----------------|------------------|
> | Baseline       | 8.08            | 4.48            | 46.3             |
> | OT only        | 8.09            | 4.76            | 47.6             |
> | TWISTED (ours) | 8.19            | 4.78            | 48.2             |
>
> ---
> > Weakness 2: The authors should provide a detailed breakdown of the OT's cost
>
> We have updated Appendix H to include more detailed computational cost analysis for OT (Table 14 above). As noted, OT only adds 2.8% in end-to-end running time. Regarding scalability, the Sinkhorn solver scales quadratically with the number of tokens (Section 2.3).
>
> ---
> > Weakness 3: The OT formulation uses fixed cost coefficients (c_d, c_w) and it is not discussed how they are selected and how sensitive are the results.
>
> We performed a hyperparameter sweep over distance and wildcard cost coefficients. We have added tables showing the sweep results in Appendix D. Note that the return does not vary greatly for different distance and wildcard costs, so optimal transport-based decoding is quite robust to the choice of distance and wildcard cost.
>
> \
> *Table 9: Average returns and scores with respect to $c_d$, a coefficient of cost for distance.*
>
> | $c_d$ | Return (\%) | Score (\%) |
> |-------|-------------|------------|
> | 0.0   | 72.08       | 32.89      |
> | 0.3   | 71.10       | 31.41      |
> | 0.6   | 72.46       | 35.60      |
> | 0.8   | 71.49       | 33.54      |
>
> \
> *Table 10: Average returns and scores with respect to $c_w$, a constant penalty for using a wildcard token.*
>
> | $c_w$ | Return (\%) | Score (\%) |
> |-------|-------------|------------|
> | 0.3   | 72.46       | 35.60      |
> | 0.6   | 71.60       | 35.41      |

---

> > ### Comment · Reviewer_tLB5 · 2025-11-27
> >
> > Thank you for the clarifications and including new testing results. My conerns have been addressed. I will keep my score.

---

### Official Review · Reviewer_oAT3 · 2025-10-30

**Soundness:** 3
**Presentation:** 3
**Contribution:** 4
**Rating:** 2
**Confidence:** 4

**Summary:**

The paper introduces a transformer-based world model for visual reinforcement learning that integrates two components: a 3D spatio-temporal positional encoding (STPE) to better capture spatial and temporal relationships across visual tokens, and an Optimal Transport (OT) decoding mechanism that formulates next-frame prediction as a transport matching problem between previous and predicted tokens. Evaluated on Craftax-Classic, Craftax, and MinAtar benchmarks, TWISTED achieves higher prediction accuracy and improved returns compared to prior transformer world models, showing faster convergence and more consistent visual rollouts across time.

**Strengths:**

1. The paper presents a novel perspective on generation and decoding through the integration of optimal transport, meanwhile offering a workable inductive bias that aligns well with the structured dynamics of grid-based visual environments.

2. The proposed spatio-temporal RoPE extension is well-motivated and effectively enhances temporal and spatial consistency, leading to tangible performance improvements over standard transformer encodings.

3. The paper features well-prepared figures and visualizations, which clearly convey both the model architecture and the qualitative impact of the proposed methods.

**Weaknesses:**

1. The method is evaluated only on specialized 2D grid-based benchmarks, and it remains unclear whether TWISTED can generalize to more complex or unstructured visual environments. The paper states that it targets “visual RL environments,” but the current setup does not demonstrate generalization to tasks such as 3D worlds, camera motion, or non-grid visual inputs (e.g., Minecraft/DMControl/Original Atari).

2. Within the evaluated domain, the contribution of the OT decoding component appears secondary to that of the spatio-temporal positional encoding (STPE). While OT improves temporal coherence, it incurs significant implementation, and the performance gains reported seem relatively modest compared to the added complexity.

3. The proposed approach builds upon existing transformer world model frameworks with incremental extensions. While these components are technically sound, their performance improvements appear proportional to their incremental nature, resulting in marginal gains that align with expectations given the added complexity. Consequently, the overall contribution and impact are somewhat limited in scope.

**Questions:**

1. Have the authors evaluated the model under different values of the distance cost parameter in the OT decoding step? How does this affect both prediction accuracy and computational efficiency? Why was the maximum feasible motion threshold set to four (i.e., displacement ≤ 2 cells per axis)? Was this empirically tuned, or is it based on environment-specific priors?

2. Have the authors tested environments that do not conform to the grid-based or bounded-motion assumptions? If not, how might the method handle these cases?

---

> ### Author Response · Authors · 2025-11-21
>
> Thank you very much for your thoughts and your feedback. We’re glad that you appreciate spatio-temporal positional encoding as well-motivated and effective, and optimal transport-based decoding as well-motivated and novel.
>
> ---
> > Weakness 1: The method is evaluated only on specialized 2D grid-based benchmarks
>
> We have reworded the paper to specify that it focuses on “visual grid-based RL environments”. This is in line with prior work on transformer world models in grid-based environments [1].
>
> ---
> > Weakness 2: While OT improves temporal coherence, it incurs significant implementation, and the performance gains reported seem relatively modest compared to the added complexity.
>
> OT does not add significant computational complexity; it increases total running time by only 2.8% (Table 14 in Appendix H). The performance gains made by OT are statistically significant, as shown in the ablations in Table 2.
>
> \
> *Table 14: Running times on a single Nvidia RTX 3090 GPU.
> WM training measures one epoch of world model training. Imagination measures one epoch of policy training in imagination. Total time represents end-to-end training time for 1M environment steps.*
>
> | Method         | WM training (m) | Imagination (m) | Total time (hrs) |
> |----------------|-----------------|-----------------|------------------|
> | Baseline       | 8.08            | 4.48            | 46.3             |
> | OT only        | 8.09            | 4.76            | 47.6             |
> | TWISTED (ours) | 8.19            | 4.78            | 48.2             |
>
> \
> *Table 2: Ablations on Craftax-classic with 1M interactions.*
>
> | Method                 | Return (\%)          | Score (\%)           |
> |------------------------|----------------------|----------------------|
> | Dedieu et al. (2025)   | 68.55 $\pm$ 0.72     | 27.24 $\pm$ 0.86     |
> | Relative PE only       | 69.26 $\pm$ 0.50     | 29.37 $\pm$ 0.80     |
> | STPE only              | 71.85 $\pm$ 0.63     | 33.94 $\pm$ 1.10     |
> | Optimal transport only | 69.77 $\pm$ 0.65     | 31.08 $\pm$ 0.88     |
> | TWISTED (ours)          | **72.46** $\pm$ 0.45 | **35.60** $\pm$ 0.92 |
>
> ---
> > Weakness 3: The proposed approach builds upon existing transformer world model frameworks with incremental extensions.
>
> The state-of-the-art performance of TWISTED is statistically significant across 3 different benchmarks. Comprehensive experiments show these improvements are robust and more than incremental:
>
> \
> *Excerpt of Table 1: Results on Craftax-classic after 0.5M and 1M environment interactions.*
>
> | Method                            | Return @ 0.5M (\%)               | Score @ 0.5M (\%)                | Return @ 1.0M  (\%)               | Score @ 1.0M (\%)                 |
> |-----------------------------------|---------------------------|---------------------------|---------------------------|----------------------------|
> | Dedieu et al. (2025) (reproduced) | 48.17 $\pm$ 0.82          | 10.22 $\pm$ 0.20          | 68.14 $\pm$ 0.42          | 24.89 $\pm$ 0.74           |
> | TWISTED (ours)                    | **63.10** $\pm$ 1.24 | **20.12** $\pm$ 0.80 | **72.46** $\pm$ 0.45 | **35.60** $\pm$  0.92 |
>
> \
> *Table 4: Results on Craftax after 1M environment interactions.*
> | Method                      | Return (\%)              | Score (\%)               |
> |-----------------------------|--------------------------|--------------------------|
> | Dedieu et al. (2025) | 5.44 $\pm$ 0.25          | 1.53 $\pm$ 0.10          |
> | Simulus                     | 6.59                     | -                      |
> | TWISTED (ours)     | **7.09** $\pm$ 0.20 | **2.40** $\pm$ 0.04 |
>
> \
> *Table 5: Returns on MinAtar after 1M environment interactions.*
> | Method                      | Asterix                   | Breakout                  | Freeway                   | SpaceInvaders              |
> |-----------------------------|---------------------------|---------------------------|---------------------------|----------------------------|
> | AD (Guan et al., 2023) | 21.05 $\pm$ 0.65          | 27.78 $\pm$ 0.16          | 57.68 $\pm$ 0.07          | 140.36 $\pm$ 1.70          |
> | Dedieu et al. (2025) | 44.81 $\pm$ 3.54          | 93.92 $\pm$ 1.44          | 71.12 $\pm$ 0.13          | 186.16 $\pm$ 1.25          |
> | TWISTED (ours)     | **50.04** $\pm$ 2.98 | **99.53** $\pm$ 2.31 | **71.34** $\pm$ 0.07 | **188.85** $\pm$ 0.62 |

---

> ### Author Response · Authors · 2025-11-21
>
> > Question 1: Have the authors evaluated the model under different values of the distance cost parameter in the OT decoding step? ... Why was the maximum feasible motion threshold set to four (i.e., displacement ≤ 2 cells per axis)?
>
> Yes, we evaluated TWISTED under different distance costs for OT decoding. We have added results over different distance costs and wildcard costs to Appendix D (Tables 9 and 10). Overall, OT decoding is robust to different choices of distance and wildcard cost; return does not differ greatly. The maximum feasible motion threshold was set to **2** (squared L2 distance $\leq$ 4) based on environment-specific priors; in Craftax, an object can move 1 position due to the player’s camera movement, and if the object is a creature, it can move an additional position in any direction.
>
> \
> *Table 9: Average returns and scores with respect to $c_d$, a coefficient of cost for distance.*
>
> | $c_d$ | Return (\%) | Score (\%) |
> |-------|-------------|------------|
> | 0.0   | 72.08       | 32.89      |
> | 0.3   | 71.10       | 31.41      |
> | 0.6   | 72.46       | 35.60      |
> | 0.8   | 71.49       | 33.54      |
>
> \
> *Table 10: Average returns and scores with respect to $c_w$, a constant penalty for using a wildcard token.*
>
> | $c_w$ | Return (\%) | Score (\%) |
> |-------|-------------|------------|
> | 0.3   | 72.46       | 35.60      |
> | 0.6   | 71.60       | 35.41      |
>
> ---
> > Question 2: Have the authors tested environments that do not conform to the grid-based or bounded-motion assumptions?
>
> Our paper’s focus is on popular grid-based environments such as Craftax, in line with prior work on transformer world models [1]. However, our method might be extended to non-grid environments by discretizing image inputs into a grid-like structure e.g. via VQ-VAE.
>
> ---
> [1] Dedieu et al. “Improving Transformer World Models for Data-Efficient RL.” ICML, 2025.

---

> > ### Comment · Reviewer_oAT3 · 2025-11-25
> >
> > I appreciate the authors' efforts in the rebuttal. Regarding weakness 2, my point is not about runtime overhead, which I understand is minimal. My concern is that the OT component introduces additional system-level complexity and potential instability. For users who may want to integrate TWISTED into larger pipelines or production systems, this added complexity and marginal performance gain could make OT less appealing in practice, and it is possible that only the spatial-temporal positional encoding would ultimately be retained.
> >
> > My primary concern remains the grid-based assumption. It is unclear how many real-world tasks can be naturally expressed as grid-based environments, and therefore how broadly applicable the proposed method truly is. While the authors mention the possibility of extending TWISTED to non-grid domains using VQ-VAE, earlier work such as IRIS has already adopted VQ-VAE and does not rely on grid constraints. The statement that TWISTED might be extended is not sufficient evidence that the method can in fact work in more general environments. Therefore I will be maintaining my current score for now.

---

> ### Author Response · Authors · 2025-12-03
>
> > Weakness 1: The method is evaluated only on specialized 2D grid-based benchmarks
> >
> > Question 2: Have the authors tested environments that do not conform to the grid-based or bounded-motion assumptions?
>
> Thank you for suggesting additional non-grid environments to demonstrate the generality of TWISTED and strengthen our paper.
>
> We have added experiments demonstrating state-of-the-art performance on Atari 100K (Section 4.4), a specific setting of Original Atari with non-grid visual inputs. These additional experiments show that TWISTED can generalize to more complex and unstructured visual environments.
>
> \
> *Table 6: Aggregate metrics on Atari 100K after 100K environment interactions. Return for each game is evaluated on 100 evaluation episodes at the end of training.*
> |                      | IQM (↑) | Optimality Gap (↓) | Mean (↑) | Median(↑) |
> |----------------------|---------|--------------------|----------|-----------|
> | Simulus (reproduced) |   0.969 |              0.410 |    **1.636** |     0.739 |
> | TWISTED (ours)       |   **1.092** |              **0.376** |    1.616 |     **0.978** |

---

### Author Response · Authors · 2025-11-21

We’d like to thank all the reviewers for their time and feedback. We appreciate that the reviewers find spatio-temporal positional encoding well-motivated and effective, optimal-transport decoding novel and intuitive, and the experiments comprehensive and convincing with state-of-the-art performance across 3 benchmarks.

In response to reviewers’ suggestions, we have added additional experiments and revised the paper (edits are shown in blue text). In particular,
* To further show the effectiveness of optimal transport-based decoding in exploiting visual redundancy, we have added an experiment comparing OT to $\Delta$-IRIS[1], another approach that addresses visual redundancy (Table 13 in Appendix G). Our experiment of adding the $\Delta$-IRIS method to TWISTED with OT disabled shows that OT outperforms $\Delta$-IRIS. This demonstrates that OT is more successful than previous methods in exploiting redundancy between frames.

* We have provided additional data on the computational costs of optimal transport-based decoding (OT) after optimizing the code (Table 14 in Appendix H). Notably, OT only increases the end-to-end training time by 2.8%, demonstrating that it adds minimal computational complexity.

\
*Table 13: Comparison of applying $\Delta$-IRIS to TWISTED without OT, on Craftax-classic after 1M environment interactions.*
| Method                           | Return (\%)          | Score (\%) |
|----------------------------------|----------------------|----------------------|
| TWISTED without OT + $\Delta$-IRIS | 28.77 $\pm$ 0.67 | 5.84 $\pm$ 0.31   |
| TWISTED (ours)                   | **72.46** $\pm$ 0.45 | **35.60** $\pm$ 0.92 |

\
*Table 14: Running times on a single Nvidia RTX 3090 GPU. WM training measures one epoch of world model training. Imagination measures one epoch of policy training in imagination. Total time represents end-to-end training time for 1M environment steps.*

| Method         | WM training (m) | Imagination (m) | Total time (hrs) |
|----------------|-----------------|-----------------|------------------|
| Baseline       | 8.08            | 4.48            | 46.3             |
| OT only        | 8.09            | 4.76            | 47.6             |
| TWISTED (ours) | 8.19            | 4.78            | 48.2             |

---
[1] Micheli et al. “Efficient World Models with Context-Aware Tokenization.” ICML, 2024.

---

### Author Response · Authors · 2025-12-03

We thank the reviewers for suggesting experiments in additional environments that strengthen our paper by demonstrating generality across environments.

We have added additional experiments on the Atari 100K environment to evaluate TWISTED on non-grid environments (Section 4.4). For Atari 100K, we applied our method on the state-of-the-art token-based world model Simulus[1], since the Dedieu et al. baseline[2] is not designed for or tested on Atari 100K. We significantly outperform the baseline in interquartile mean (IQM), optimality gap, and median across the 26 games of the benchmark. IQM and optimality gap are the most important metrics for Atari 100K, because they are more robust to outliers and more statistically consistent than mean and median [3]. By achieving new state-of-the-art performance for Atari 100K, TWISTED demonstrates its generality across visual RL environments, including non-grid environments.

\
*Table 6: Aggregate metrics on Atari 100K after 100K environment interactions. Return for each game is evaluated on 100 evaluation episodes at the end of training.*
|                      | IQM (↑) | Optimality Gap (↓) | Mean (↑) | Median(↑) |
|----------------------|---------|--------------------|----------|-----------|
| Simulus (reproduced) |   0.969 |              0.410 |    **1.636** |     0.739 |
| TWISTED (ours)       |   **1.092** |              **0.376** |    1.616 |     **0.978** |

---
[1] Cohen et al. “Uncovering Untapped Potential in Sample-Efficient World Model Agents”. arXiv preprint arXiv:2502.11537, 2025.

[2] Dedieu et al. “Improving Transformer World Models for Data-Efficient RL.” ICML, 2025.

[3] Agarwal et al. “Deep Reinforcement Learning at the Edge of the Statistical Precipice.” NeurIPS, 2021.

---

### Author Response · Authors · 2025-12-03
**Author Final Remarks**

Dear Reviewers and Area Chair,

First, we’d like to thank all the reviewers for their detailed comments. Your feedback has helped us strengthen the paper during the rebuttal period.

Overall, we are happy that reviewers found that
* TWISTED’s spatio-temporal encoding is “well-motivated and effectively enhances temporal and spatial consistency”
* TWISTED’s optimal transport-based decoding is “both novel and intuitive”, “original”, and “novel and well-motivated”
* The experiments “are informative and well executed”, “show consistent improvement across datasets”, “outperforms previous baselines”, and are “comprehensive and convincing, demonstrating SOTA performance”

We have addressed every weakness and question raised by reviewers, through modifications to the paper and additional experiments requested by reviewers. In particular,
* Reviewers oAT3 and Niit were concerned that TWISTED was limited to grid-based and visually redundant environments. To address this, we added experiments on an additional environment (Atari 100K), **demonstrating that TWISTED’s superior performance is not limited to grid-based environments**.
* Reviewers oAT3, tLB5, Niit, and p88M expressed concern about optimal transport-based decoding’s (OT) computational complexity and cost. To address this, we added additional data on OT’s computational cost, showing that **OT only increases end-to-end computation time by 2.8%**.
* Reviewer Niit expressed concern that the OT component by itself may not outperform previous solutions to visual redundancy between frames, such as $\Delta$-IRIS[1]. To address this, we added an additional experiment that compared OT directly to $\Delta$-IRIS, **demonstrating that OT outperforms $\Delta$-IRIS**.

Given the reviewers’ confidence in the paper’s empirical evaluation and our comprehensive response to reviewers’ concerns, we hope this paper will receive careful consideration for acceptance.

---
[1] Micheli et al. “Efficient World Models with Context-Aware Tokenization.” ICML, 2024.

---

### Meta-Review · Area_Chair_3wnE · 2026-01-04

**Summary:**

This work builds upon existing transformer world models by further exploiting spatial-temporal relationships between visually adjacent tokens for visual environment modeling. The proposed solution, TWISTED, augments transformer world models with a 3D spatio-temporal positional encoding (STPE) and an optimal-transport (OT) based decoding strategy. Experiments show promising performances on the Craftax-classic, Craftax, MinAtar, and Atari 100K benchmarks. Reviewers' opinions are at odd for this work. On the positive side, the strong empirical performances and the insights into spatio-temporal relationship modeling are favored. On the negative side, the proposed STPE overlaps substantially with existing 3D RoPE (in video generation models, world models etc), while the OT-based decoding scheme makes the solution more complex and less pluggable into other world models. The AC finds that the paper appears borderline in its current form.

**Reviewer Concerns:**

Authors have performed an effective rebuttal, which has addressed many concerns raised in the preliminary reviews. The biggest concern still remaining is: The 3D spatio-temporal positional encoding (STPE) is a simple extension of 3D RoPE with action encodings, while the paper would need a reshape to highlight its specific new contribution in parallel to 3D RoPE. This issue warrants a rejection of the paper in its current form.

**Reviewer Scores:**

Reviewer oAT3: Most concerns addressed. Based on the final notes of the reviewer, the score may be increased from 2 to 4 at worst and 6 at best.
Reviewer Niit: The biggest concern remains. The score will be changed from 2 to 4 at best, but still challenges the acceptance.
The rest of reviewers will keep their positive scores.

---

### Decision · Program_Chairs · 2026-01-26

Reject